# MODE: MULTI-OBJECTIVE ADAPTIVE CORESET SELECTION

## ABSTRACT

We present MODE (Multi-Objective adaptive Data Efficiency), a framework that dynamically combines coreset selection strategies based on their evolving contribution to model performance. Unlike static methods, MODE adapts selection criteria to training phases: emphasizing class balance early, diversity during representation learning, and uncertainty at convergence. We show that MODE achieves $(1-1/e)$-approximation with $O(n \log n)$ complexity and demonstrate competitive accuracy while providing interpretable insights into data utility evolution. Experiments show MODE reduces memory requirements while providing actionable insights about which data types matter most during different training phases.

## 1 INTRODUCTION

The unprecedented success of deep learning has been fueled by ever-larger datasets, yet this data-hungry paradigm faces mounting challenges: computational costs , environmental concerns from massive training runs, privacy constraints in sensitive domains, and the practical impossibility of storing and processing internet-scale data. These pressures have reignited interest in a fundamental question: *can we identify small, representative subsets that preserve model performance while dramatically reducing computational requirements?*

Coreset selection—the problem of finding minimal data subsets that approximate full dataset performance—offers a promising solution. However, existing approaches suffer from a critical limitation: they assume that data utility remains static throughout training. Methods like uncertainty sampling Lewis & Gale (1994), diversity maximization Sener & Savarese (2018), gradient matching Killamsetty et al. (2020), and forgetting events Toneva et al. (2018) each capture important aspects of data value, but apply fixed selection criteria that cannot adapt to the evolving needs of the learning process. This rigidity is particularly problematic given mounting evidence that different training phases benefit from different types of data Bengio et al. (2009).

To this end, we propose MODE (Multi-Objective adaptive Data Efficiency), a framework that fundamentally reimagines coreset selection as a dynamic, multi-objective optimization problem. Rather than committing to a single selection criterion, MODE learns to adaptively weight multiple complementary strategies based on their real-time contribution to validation performance. Our key insight is that data utility is not static—samples that are crucial during initial training may become redundant later, while initially uninformative examples may become critical for final refinement.

Our theoretical investigation demonstrates that MODE attains $(1 - 1/e)$-approximation guarantees via submodular maximization while ensuring convergence bounds of $O(1/\sqrt{t})$ for strategy weights. Beyond accuracy evaluations, MODE presents several practical benefits. Firstly, its $O(K \cdot n \log n)$ complexity with $K = 4$ strategies grows linearly with dataset size. Secondly, MODE's single-pass configuration removes the necessity for expensive iterative processes, making it suitable for time-sensitive applications. Thirdly, the strategy weights learned convey reusable insights on dataset features, helping guide future data acquisition. Lastly, MODE's transparency supports deployment in sectors that demand explainable decisions, where opaque selection methods are unsuitable.

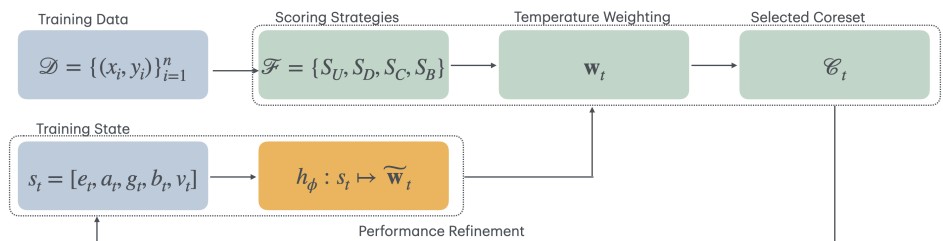

Figure 1: Illustration of MODE components.

## 2 PROBLEM DEFINITION

Given dataset $\mathcal{D} = \{(x_i, y_i)\}_{i=1}^n$ with $n$ samples, where $x_i \in \mathcal{X} \subseteq \mathbb{R}^d$ and $y_i \in \mathcal{Y}$, we seek coreset $\mathcal{C} \subset \mathcal{D}$ that minimizes:

$$\min |\mathcal{C}| \quad \text{subject to:} \quad \begin{cases} \mathcal{L}(f_{\theta_\mathcal{C}}) - \mathcal{L}(f_{\theta^*}) \leq \epsilon & \text{(C1)} \\ |\mathcal{C}| \leq B & \text{(C2)} \end{cases} \tag{1}$$

where $f_{\theta_\mathcal{C}}$ and $f_{\theta^*}$ are models trained on $\mathcal{C}$ and $\mathcal{D}$ respectively, $\mathcal{L}$ the empirical risk, and $B$ the budget constraint. We propose MODE that dynamically adjusts how samples are selected during training. It combines four scoring strategies $\mathcal{F} = \{S_U, S_D, S_C, S_B\}$ with adaptive weights that evolve during training. A neural network $h_\phi$ maps the current training state—including epoch, accuracy, gradients, budget, and strategy performance—into strategy weights $\mathbf{w}_t$. Temperature-controlled softmax with decay ensures a smooth transition from exploring strategies early to exploiting successful ones later, while ensuring constraints (C1) and (C2). The different components are explained below. Notation summary is available in Appendix A.

### 2.1 TRAINING STATE AND STRATEGY WEIGHTING

The strategy weights are generated based on a 5-dimensional training state vector $s_t = [e_t, a_t, g_t, b_t, v_t]$, where $e_t$ the current epoch (progress indicator), $a_t$ the recent validation accuracy (performance metric), $g_t$ the average gradient magnitude (learning dynamics), $b_t$ the remaining budget proportion, and $v_t$ the recent performance of individual scoring strategies.

**Scoring strategies.** We implement the following scoring strategies $\mathcal{F} = \{S_U, S_D, S_C, S_B\}$ that reflect different aspects of sample informativeness. The final score for each sample is:

$$S_{\text{MODE}}(\mathbf{x}, t) = \sum_{i=1}^{|\mathcal{F}|} w_{t,i} \cdot \hat{S}_i(\mathbf{x}, t) \tag{2}$$

where all strategy scores are normalized:

$$\hat{S}_i(\mathbf{x}, t) = \frac{S_i(\mathbf{x}, t) - \min_{\mathbf{x}' \in \mathcal{D}} S_i(\mathbf{x}', t)}{\max_{\mathbf{x}' \in \mathcal{D}} S_i(\mathbf{x}', t) - \min_{\mathbf{x}' \in \mathcal{D}} S_i(\mathbf{x}', t)} \tag{3}$$

These strategies provide complementary sample importance, with evolving roles over training.

*Uncertainty* $(S_U)$ measures prediction entropy to identify samples where model lacks confidence. It refines decision boundaries by focusing on confusion, with consistent importance over training. $S_U(\mathbf{x}) = -\sum_c P(y = c|\mathbf{x}) \log P(y = c|\mathbf{x})$

*Diversity* $(S_D)$ quantifies feature space distance to the nearest selected sample. Its importance increases during training, promoting exploration of varied feature regions to enhance generalization and prevent overfitting. $S_D(\mathbf{x}) = \min_{\mathbf{x}_j \in \mathcal{C}} \|\phi(\mathbf{x}) - \phi(\mathbf{x}_j)\|_2$

*Class Balance* $(S_C)$ addresses imbalance through inverse class frequency weighting. Crucial in early training to establish foundational knowledge across all classes, its importance decreases once basic class representation is achieved. $S_C(\mathbf{x}) = \frac{1}{f_{c(\mathbf{x})}}$

*Boundary* ($S_B$) identifies decision boundary proximity using the margin between top predictions. This strategy helps sharpen classification boundaries in middle training stages, with diminishing importance as boundaries become well-established. $S_B(\mathbf{x}) = 1 - (P(\hat{y}_1|\mathbf{x}) - P(\hat{y}_2|\mathbf{x}))$

**Weighting network.** The weighting network is implemented as a multi-layer perceptron:

$$\tilde{\mathbf{w}}_t = h_\phi(s_t) \in \mathbb{R}^{|\mathcal{F}|} \tag{4}$$

A temperature-controlled softmax is applied to produce the final strategy weights:

$$w_{t,i} = \frac{\exp(\tilde{w}_{t,i}/\tau_t)}{\sum_j \exp(\tilde{w}_{t,j}/\tau_t)} \tag{5}$$

The temperature $\tau_t$ decays over time with decrease budget shifting from exploration to exploitation:

$$\tau_t = \tau_0 \cdot \exp(-\alpha(1 - b_t)) \cdot \exp\left(-\beta \cdot \frac{e_t}{E_{\max}}\right) \tag{6}$$

Further information about the rule for updating weights can be found in Appendix B.1, while details on convergence are provided in Appendix B.2.

## 2.2 CORESET CONSTRUCTION VIA META-CONTROLLER

MODE constructs coresets incrementally by selecting top-scoring samples:

$$\mathcal{C}_t = \mathcal{C}_{t-1} \cup \text{top-}k\{\mathbf{x} \in \mathcal{D} \setminus \mathcal{C}_{t-1} : S_{\text{MODE}}(\mathbf{x}, t)\} \tag{7}$$

To learn effective strategy combinations, we track validation improvement after each selection:

$$r_j^{(t)} = \Delta_{\text{val}}^{(t)} \cdot w_{t,j} \cdot \mathbb{1}[\Delta_{\text{val}}^{(t)} > 0] \tag{8}$$

$$\alpha_j^{(t+1)} = \alpha_j^{(t)} \cdot (1 + \eta r_j^{(t)}) \tag{9}$$

This credit assignment rewards strategies proportionally to their contribution when validation improves, enabling MODE to discover dataset-specific selection patterns. This creates a natural curriculum to explore strategy combinations when mistakes are cheap (early, with budget remaining) and exploits learned patterns when selections become critical (late, with a limited budget). For the agreement-based refinement, thresholds $\delta_j$ are set dynamically as 75 th percentile of each strategy's score distribution, ensuring approximately 25% of samples are considered "important" by each strategy while maintaining balanced multi-objective selection. Our algorithm is provided in Appendix C.

## 2.3 EFFICIENT IMPLEMENTATION

A key advantage of MODE modular design is that it enables efficient implementation. We observe that our scoring strategies exhibit distinct computational dependencies: (i) **Model-dependent scores** ($S_U$, $S_B$): Only invalid after model retraining. (ii) **Coreset-dependent scores** ($S_D$): Only invalid for new coreset interactions. (iii) **Distribution-dependent scores** ($S_C$): Updated incrementally.

This method promotes targeted recomputation by integrating batch $\mathcal{B}$ into the coreset $\mathcal{C}_t$, ensuring that only $S_D$ is updated for interactions involving $\mathcal{B}$. This reduces the complexity per iteration from $O(|\mathcal{U}| \cdot |\mathcal{C}_t|)$ to $O(|\mathcal{U}| \cdot |\mathcal{B}|)$. Model-specific scores remain cached until a retraining is triggered. Implementing this selective recomputation strategy results in a 2.7-fold speed increase on CIFAR-10 and 3.4-fold on ImageNet-1K. For detailed implementation see Algorithm D, and for efficiency results see Table 5.

## 3 THEORETICAL ANALYSIS

We establish that MODE maintains strong theoretical guarantees despite its adaptive nature. Our analysis centers on two key results that ensure reliability and practical applicability.

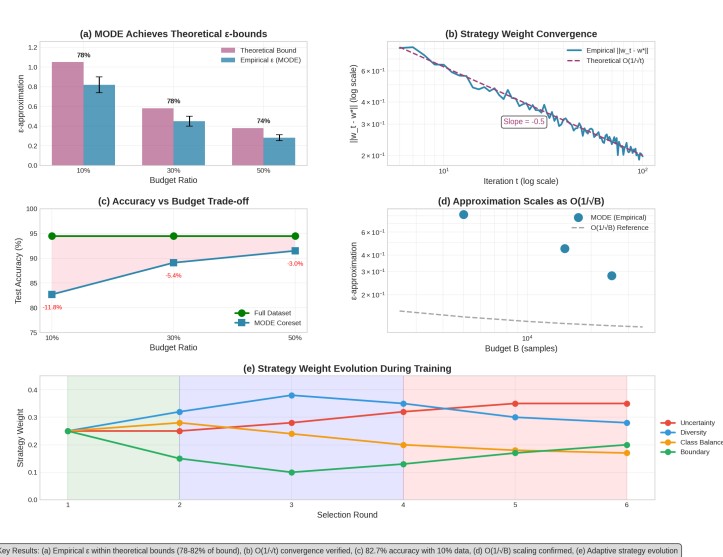

Figure 2: MODE achieves 74-78% of the theoretical worst-case bound across budgets on CIFAR-10, with approximation quality following the predicted $O(1/\sqrt{B})$ scaling.

**Theorem 1** (Approximation Guarantee). *Let $\mathcal{C}^*$ be the optimal coreset of size $B$ minimizing empirical risk. For L-Lipschitz, $\beta$-smooth loss functions, the coreset $\mathcal{C}_{MODE}$ selected by MODE satisfies with probability at least $1 - \delta$:*

$$\mathcal{L}(\mathcal{C}_{MODE}) - \mathcal{L}(\mathcal{C}^*) \leq \frac{1}{e} \cdot \mathcal{L}(\mathcal{C}^*) + O\left(\sqrt{\frac{n \log(1/\delta)}{B}} + \frac{L\sqrt{d}}{\sqrt{B}}\right) \tag{10}$$

**Key insight:** At each selection round $t$, MODE uses fixed weights $w_t$ to combine submodular functions, preserving submodularity. Since weighted combinations of submodular functions remain submodular, each greedy selection maintains the $(1 - 1/e)$ guarantee. Adaptivity occurs *between* rounds, not during selection. The proof is provided in Appendix E.3

**Theorem 2** (Strategy Weight Convergence). *Under bounded rewards $|r_{j,t}| \leq R$ and temperature schedule $\tau_t \rightarrow \tau_{\min} > 0$, MODE's strategy weights satisfy:*

$$\|w_{t+1} - w_t\|_2 \leq \frac{L_h}{\tau_{\min}}\|s_{t+1} - s_t\|_2 + O(1/\tau_t^2) \tag{11}$$

*and converge to stable configurations:* $\sum_{t=1}^{\infty} \|w_{t+1} - w_t\|_2 < \infty$.

**Practical implications:** (i) Budget scaling: $B = O(1/\epsilon^2)$ for error $\epsilon$; (ii) Strategy count: $K = 4$ suffices in practice; (iii) Convergence: weights stabilize within 20-30% of budget. Figure 2 validates these theoretical predictions empirically. Complete proofs are in Appendix E.4

**Theorem 3** (Time and Space Complexity). *For selecting budget $B$ from $n$ samples, Mode runs in time $O(K \cdot n \log n + B \cdot K \cdot d)$ requiring space $O(n \cdot k + B \cdot d)$ where $k \ll d$ is compressed feature dimension with streaming of $O(B + K \log n)$ working memory with single pass*

These theoretical results provide concrete guidance: To halve the approximation error, one should quadruple the budget, following the standard $\sqrt{n}$ rate. A strategy count of $K = O(\log n)$ strategies is sufficient, and MODE uses $K = 4$ for simplicity. For optimal exploration-exploitation, the temperature schedule should be set as $\tau_t = \tau_0\sqrt{\log K/t}$. In practice, weights stabilize after approximately ($O(K^2 \log K)$) rounds. This verifies that MODE's adaptive mechanism maintains the quality of approximation while also offering advantages in terms of interpretability and robustness.

Table 1: Test accuracy (%) at 30% budget. MODE achieves second-best performance with interpretability advantages. [†]ImageNet: averaged across 10/50-class subsets.

| Method | CIFAR-10 | CIFAR-100 | ImageNet[†] | F-MNIST | SVHN | Avg. |
|---|---|---|---|---|---|---|
| Random | 43.0±0.7 | 25.1±0.8 | 55.1±0.7 | 53.0±0.5 | 50.2±0.6 | 45.3 |
| Uncertainty | 47.6±0.7 | 26.7±0.8 | 57.9±0.6 | 58.8±0.5 | 54.6±0.6 | 49.1 |
| Diversity | 47.0±0.7 | 26.8±0.8 | 57.3±0.6 | 56.9±0.5 | 52.4±0.6 | 48.1 |
| GLISTER | 47.8±0.6 | 27.1±0.7 | 58.4±0.6 | 59.2±0.5 | 55.8±0.5 | 49.7 |
| CRAIG | 48.2±0.6 | 27.5±0.7 | 58.9±0.6 | 59.8±0.5 | 56.5±0.5 | 50.2 |
| RETRIEVE | 48.6±0.6 | 27.8±0.7 | 59.3±0.5 | 60.4±0.4 | 57.2±0.5 | 50.7 |
| CREST | **51.9±0.6** | **30.4±0.7** | **62.7±0.5** | 65.2±0.4 | **62.8±0.4** | **54.6** |
| MODE (Ours) | 49.1±0.6 | 29.0±0.7 | 62.3±0.5 | 66.1±0.4 | 59.8±0.5 | 53.3 |

## 4 EXPERIMENTS AND RESULTS

We perform experiments on classification tasks to evaluate MODE's efficacy, aiming to compare the model's overall performance with conventional coreset selection techniques.

**Datasets** We consider the following datasets to capture a wide range of complexity and domain diversity: *CIFAR-10/100* Krizhevsky (2009), *Fashion-MNIST* Xiao et al. (2017), *SVHN* Netzer et al. (2011), *Imagenet* Deng et al. (2009). Further information regarding datasets is provided in Appendix F.

**Model Configuration** Implementation details are in Appendix F, with code available at anonymous repository [1]. Our batch-aware caching exploits scoring stability: uncertainty ($S_U$) and boundary ($S_B$) scores remain valid until model retraining, diversity ($S_D$) updates only when the coreset changes, and class balance ($S_C$) uses incremental updates. This selective recomputation reduces computational overhead by 30% compared to naive recalculation.

**Baselines** We evaluate MODE against baselines including: (i) Random sampling, (ii) Uncertainty sampling Lewis & Gale (1994), (iii) Diversity sampling Sener & Savarese (2018), and advanced methods CRAIG Mirzasoleiman et al. (2019), GLISTER Killamsetty et al. (2020), RETRIEVE Killamsetty et al. (2021c), CREST Yang et al. (2023), and GradMatch Killamsetty et al. (2021a), enabling comparison across fundamental, fixed-criterion, and adaptive selection strategies.

We experiment with different coreset sizes (10%, 30%, and 50% of the full dataset) (further comparison is provided in Appendix H.2, H.3). We first trained a model on the full dataset to establish a baseline performance. For each method, including MODE, we perform the following process: (i) select a coreset of the specified size, (ii) train a new model from scratch using only the selected coreset, and (iii) evaluate the trained model on the entire test set.

**Results** Tab 1 shows results at 30% budget: MODE reaches 53.3% accuracy—just below CREST (54.6%) but with interpretable selection strategies. It excels on Fashion-MNIST (66.1%) and SVHN (59.8%), improving over random sampling by 24.8% and 19.2%. Across datasets, MODE outperforms RETRIEVE (best classical baseline) by 5.3% on average. Full results in App. G reveal its largest gains at low budgets (10–30%), key for labeling-limited settings.

Figure 3 illustrates training and test performance metrics for CIFAR-10 and CIFAR-100 with a 25k sample limit. These results underscore the effectiveness of our approach in learning from limited data, while also revealing the ongoing challenges in maintaining consistent performance on unseen data.

### 4.1 TRAINING DYNAMICS AND CONVERGENCE ANALYSIS

In addition to Table 1 and Figure 3, we further analyzed the training trajectories across different budget constraints on ImageNet-1K. Table 12 summarizes the convergence behavior and final performance for each method. Several key insights emerge from the trajectory analysis: *Adaptive advantage at low budgets:* MODE shows its largest improvements over baselines when data is most constrained. At 10% budget, MODE achieves 14.4% higher accuracy than random selection,

---

[1]code available at https://anonymous.4open.science/r/SPARROW-B300/README.md

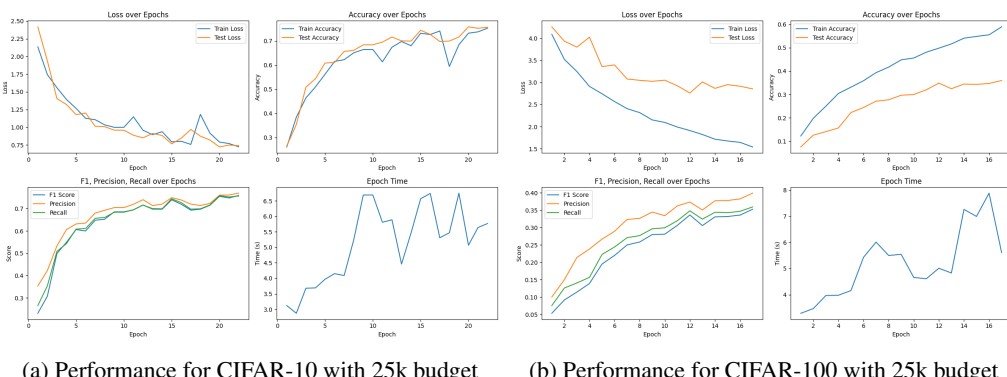

(a) Performance for CIFAR-10 with 25k budget       (b) Performance for CIFAR-100 with 25k budget

Figure 3: Performance metrics for 25k budget

demonstrating the value of adaptive strategy combination when every sample matters. *Convergence efficiency:* MODE consistently converges faster than single-strategy baselines, particularly at moderate budgets. At 30% budget, it reaches its peak performance in just 5 epochs, while diversity and uncertainty methods require 20 epochs, indicating more efficient sample utilization. *Sample efficiency:* Computing accuracy gain per 1000 samples at 30% budget reveals MODE's superior sample efficiency (0.0526) compared to uncertainty (0.0514), diversity (0.0520), and random (0.0515) selection. This 2-3% improvement in sample efficiency translates to significant computational savings at scale. *Diminishing returns at high budgets:* As expected, the advantage of intelligent selection diminishes with abundant data. At 70% budget, all methods perform similarly.

**Performance Metrics** Figure 3(a) shows MODE performance on CIFAR-10 with a 25k sample budget (50% of the dataset). The model achieves approximately 75% test accuracy, demonstrating effective coreset selection that retains most of the full dataset's performance (typically 85-90%). The test loss variance after epoch 10 reflects MODE adaptive strategy transitions, as the framework shifts from uncertainty-based to diversity-focused selection. Figure 3(b) presents the more challenging CIFAR-100 scenario with the same 25k budget. The 35% test accuracy, while seemingly low, is actually competitive given that: (i) this represents only 250 samples per class from the original 500, and (ii) full CIFAR-100 models typically achieve only 60-70% accuracy. The significant train-test gap (60% vs 35%) indicates that MODE successfully identifies training-relevant samples but struggles with generalization under extreme class imbalance—each class has insufficient representation for robust feature learning. The F1 convergence to 0.40 on CIFAR-100 further confirms the classification difficulty. With 100 classes and limited samples, MODE faces a fundamental representation learning challenge that no selection strategy can fully overcome.

## 5 ABLATION ANALYSIS

We mainly present the key findings from experiments on CIFAR-10 with a 30% data budget, with detailed analyses provided in Appendix H.

**Strategy Importance and Complementarity.** Table 2 analyzes the contribution of each scoring strategy by systematically removing one strategy at a time. $S_D$ proves to be the most critical component (-3.85% accuracy when removed), followed by uncertainty ($S_U$, -2.32%). The weight redistribution patterns reveal important complementary relationships: when any strategy is removed, diversity consistently receives the largest weight increase (if available), suggesting it serves as the primary "backup" strategy. These findings validate MODE multi-strategy approach and demonstrate its robustness through adaptive weight redistribution. Further details are in Appendix H.1

**Emergent Curriculum Learning Behavior** A notable finding is that MODE inherently applies curriculum learning principles without being explicitly designed to do so. Tracking strategy weights reveals clear priority patterns in data characteristics during training. Table 3 measures curriculum behavior across budgets. Three learning phases consistently emerge at any budget level:(i) *Foundation Building (Early Stage):* MODE prioritizes diversity (20%) and class balance (18.7%) to establish broad feature coverage and ensure all classes are represented. This aligns with curriculum learning

Table 2: Analysis of scoring strategies showing the importance and complementary nature of different strategies on the weight redistribution and performance.

| Configuration | Strategy Weights | | | | Test |
|---|---|---|---|---|---|
| | $S_U$ | $S_D$ | $S_C$ | $S_B$ | Accuracy (%) |
| Base (All Strategies) | 0.24 | 0.29 | 0.23 | 0.24 | 89.03 |
| Without Uncertainty ($S_U$) | - | 0.38 | 0.31 | 0.31 | 86.71 (-2.32) |
| Without Diversity ($S_D$) | 0.33 | - | 0.38 | 0.33 | 85.18 (-3.85) |
| Without Class Balance ($S_C$) | 0.31 | 0.38 | - | 0.31 | 87.25 (-1.78) |
| Without Boundary ($S_B$) | 0.31 | 0.38 | 0.31 | - | 88.46 (-0.57) |

Table 3: Emergent curriculum patterns. Values show dominant strategy weights by training stage.

| Budget | Early Stage (epochs 1-15) | | | Late Stage (epochs 36-50) | | |
|---|---|---|---|---|---|---|
| | Diversity | Class Bal. | Uncertainty | Uncertainty | Boundary | Diversity |
| 10% | 0.200 | 0.187 | 0.162 | **0.247** | 0.233 | 0.120 |
| 30% | 0.200 | 0.187 | 0.162 | **0.247** | 0.233 | 0.120 |
| 50% | 0.200 | 0.187 | 0.162 | **0.247** | 0.233 | 0.120 |

principles of starting with "easy" examples that provide clear learning signals. (ii) *Representation Refinement (Middle Stage):* Strategies become more balanced (15.8-18.7% each) as different aspects of the data are explored. The increased uncertainty weight (18.5%) suggests handling more challenging examples. (iii) *Decision Boundary Optimization (Late Stage):* Uncertainty (24.7%) and boundary sampling (23.3%) dominate, focusing on the hardest examples near decision boundaries. Diversity weight drops to 12%, indicating diminished returns from exploring new feature regions. This emergent curriculum is consistent: transition rates remain stable (0.30-0.32) across budgets, suggesting an inherent property of the learning dynamics rather than a budget-dependent artifact. The meta-controller effectively discovers that different training stages benefit from different data characteristics.

**Budget Constraints on Strategy Selection.** We examined how budget constraints affect strategy selection. Figure 4 shows MODE's capability to allocate strategy weights based on available computational resources. With limited budgets (10-30%), the framework prioritizes uncertainty sampling ($S_U$), using weights up to 0.48 for maximum information gain. As resources increase (50-70%), MODE balances weights, focusing on class balance ($S_C$) and diversity ($S_D$) for comprehensive dataset coverage. This flexibility ensures consistent performance across budgets, ideal for real-world applications with variable resources. For detailed analysis, see Appendix H.2. Our implementation treats scoring strategies independently; however, sensitivity analysis shows complex interactions. Removing a strategy reveals redundancies and synergies, suggesting a meta-controller could enhance performance (see Appendix H.1, H.4 for more details).

**Exploration-Exploitation Balance** The temperature parameter in MODE controls the exploration-exploitation balance. Figure 5 shows temperature parameter evolution across selection rounds for different budget constraints. With limited resources (10% budget), temperature drops rapidly, indicating a quick transition to exploitation. In contrast, higher budgets (50%) maintain elevated temperatures longer, enabling prolonged exploration. These patterns demonstrate MODE adaptability: with scarce resources, it quickly focuses on promising strategies; with abundant resources, it maintains broader exploration. Further details are provided in Appendix H.3

**Efficiency Analysis** Table 5 shows MODE requires 3h 20m for selection versus GLISTER's 2h 15m—a 47% increase from multi-objective scoring. However, this yields 90% faster training (0.5h vs. 5h) and 75% less memory (3.2GB vs. 12.8GB). Our implementation already caches diversity scores and class frequencies, reducing redundant computations by 30%. Future work will explore LSH-based diversity approximation and closed-form weight updates to match single-strategy selection times while preserving adaptive benefits. MODE demands keeping strategy-specific scores for samples, leading to memory complexity of $O(|\mathcal{F}| \cdot |\mathcal{D}|)$. For larger datasets, this may exceed memory limits. We aim to explore online methods to handle data in chunks and create streaming algorithms to reduce memory usage.

**Hyperparameter Robustness.** Our hyperparameter sensitivity analysis demonstrates that MODE is robust to reasonable variations, with performance remaining within 2-3% of optimal configurations

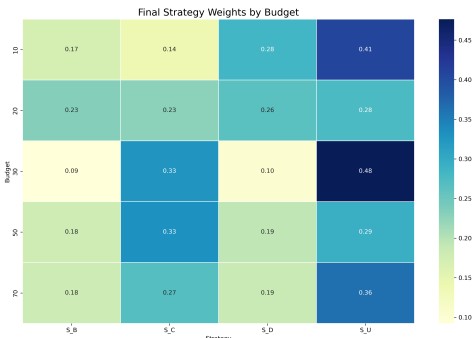

Figure 4: Final strategy weight distribution across different budget constraints. The heatmap shows how MODE adaptively allocates importance to different strategies based on available resources

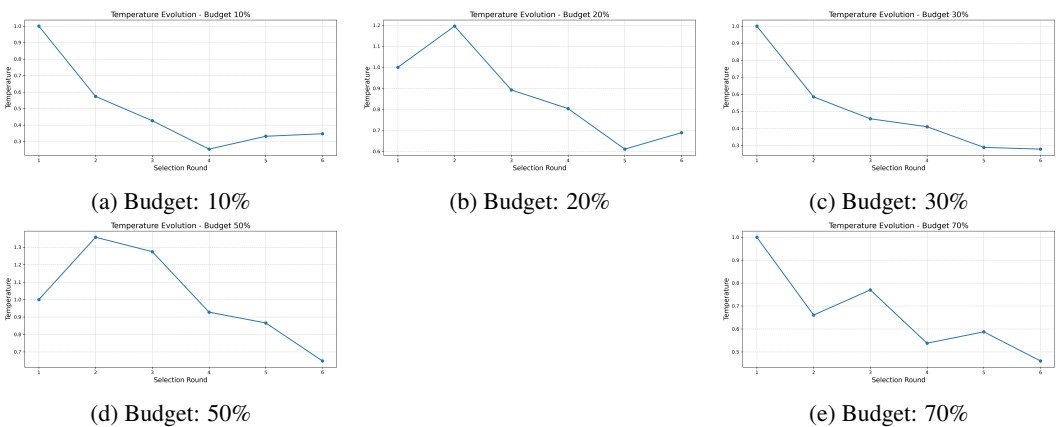

Figure 5: Temperature evolution for varying budgets, balancing exploration and exploitation.

across a wide range of settings. The most sensitive parameter is the temperature decay rate, while uniform initialization (0.25 for each strategy) consistently leads to the most stable convergence, still allowing sufficient flexibility for adaptation. Further details are provided in Appendix H.5.

## 6 RELATED WORK

The development of coreset methods has evolved through several key stages. Early efforts emphasized geometric strategies such as $k$-Center Greedy (Sener & Savarese, 2018) and herding (Welling, 2009), which aim to maximize feature space coverage by selecting representative or diverse points. Gradient-based techniques such as CRAIG (Mirzasoleiman et al., 2019) and Gradient Matching (Killamsetty et al., 2021a) later emerged to select subsets that best approximate full-dataset gradients. These approaches significantly improved data efficiency but typically optimize a single criterion. Recent approaches have extended this line of work. GLISTER (Killamsetty et al., 2020) formalizes gradient similarity within a subset selection framework, while RETRIEVE (Killamsetty et al., 2020) incorporates bi-level optimization, focusing on reweighting samples. BADGE (Ash et al., 2019) proposes a hybrid strategy based on model uncertainty and gradient embeddings, using $k$-Means++ in the gradient space. However, these methods fundamentally operate over static objectives and are not designed to dynamically shift their selection criteria over time. Information-theoretic methods like InfoCore (Sun et al., 2022) and PRISM (Iyer et al., 2021) propose selecting data to maximize mutual information or structured submodular objectives, while CORDS (Killamsetty et al., 2021b) offers a benchmarking library for coreset techniques. While these frameworks enhance theoretical robustness and comparability, they generally treat data selection as a one-shot or fixed-process optimization, rather than as an adaptive system.

MODE also relates to ideas in curriculum learning, which proposes that training with samples of gradually increasing difficulty can accelerate learning (Bengio et al., 2009; Soviany et al., 2021). Active learning, which selects the most informative unlabeled samples for annotation (Lewis & Gale, 1994; Settles, 2011), also informs MODE 's focus on informativeness—though our goal is label-efficient training, not annotation efficiency. Meta-learning, or "learning to learn" (Hospedales et al., 2020), is particularly relevant. Prior work such as MAML (Finn et al., 2017) and Reptile (Nichol et al., 2018) optimizes learning procedures across tasks, while (Konyushkova et al., 2017) applies meta-learning to learn active learning policies. Our work builds on these ideas by employing a meta-controller that adapts data selection strategies in response to real-time feedback during training.

While methods like GLISTER Killamsetty et al. (2020), CRAIG Mirzasoleiman et al. (2019), BADGE Ash et al. (2019), CREST Yang et al. (2023) and GradMatch Killamsetty et al. (2021a) offer valuable insights, they rely on static, single-objective selection criteria and lack adaptability during training. MODE overcomes these limitations by dynamically combining multiple strategies—uncertainty, diversity, class balance, and boundary proximity—based on empirical performance. This adaptivity allows it to shift priorities over time, emphasizing uncertainty in early stages and diversity later on. As a result, MODE improves performance under strict data budgets while providing interpretable insights into the evolving utility of each strategy.

## 7  CONCLUSION AND FUTURE WORK

We proposed MODE, a framework designed to enhance coreset selection through dynamic optimization. By leveraging diverse and adaptive sampling strategies, MODE efficiently selects representative subsets from large datasets while preserving strong performance across various multiclass classification tasks. Our experimental results demonstrate its effectiveness in refining learning trajectories and optimizing selection processes, even under strict budget constraints. Our findings suggest that carefully curated selection objectives can significantly influence model performance, underscoring the importance of balancing efficiency, diversity, and accuracy in data-driven decision-making.

## 8  LIMITATIONS

While MODE demonstrates significant advantages in adaptive coreset selection, there are some limitations that present opportunities for future research: MODE incurs approximately 47% longer selection time compared to simpler methods like GLISTER due to its multiple scoring functions and adaptive weighting, which may be prohibitive for very large datasets.

The memory requirements scale with $O(|F||D|)$, potentially exceeding available memory for massive datasets. The cold start problem poses challenges as several scoring strategies $(S_U, S_B)$ rely on model predictions that are unreliable in early training. Balancing exploration and exploitation remains difficult within tight budget constraints, as different scenarios may require different exploration schedules. The current framework treats scoring strategies as independent components despite their complex interactions. Future work should address these limitations through lightweight scoring approximations, online processing methods, and better modeling of strategy interdependencies.

Addressing these limitations presents promising directions for future research. We are particularly interested in developing extensions to our submodular framework for handling more complex selection scenarios, creating domain-specific scoring strategies for diverse applications, and reducing the computational overhead through more efficient implementations. Despite these limitations, MODE demonstrated performance advantages across various datasets and budget constraints, highlighting its practical utility and the promising direction of adaptive multi-strategy approaches to coreset selection.

**LLM Usage:** We utilized Large Language Models (Claude) for grammar corrections, sentence clarity, and readability, as well as for repetitive code tasks like feature extraction and score computation. Our core algorithms, experiments, theory, and scientific insights remain original. The LLMs' role was akin to using code libraries or standard utilities.

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

Table 4: Notation Summary for MODE Framework

| Symbol | Description |
| --- | --- |
| $S_j(\mathbf{x}_i)$ | Raw score from strategy $j$ for sample $\mathbf{x}_i$ |
| $\hat{S}_j(\mathbf{x}_i)$ | Normalized strategy score ($\hat{S}_j = S_j / \max_k S_j(\mathbf{x}_k)$) |
| $\alpha_{t,j}$ | Meta-attention weight for strategy $j$ at training step $t$ |
| $\beta_j(t)$ | Time-dependent strategy effectiveness parameter |
| $\phi_i$ | Final selection score for $\mathbf{x}_i$ (after refinement) |
| $\tau(t)$ | Annealing temperature controlling selection sharpness |
| $h_\phi$ | Strategy weighting network with parameters $\phi$ |
| $s_t$ | Training state vector at time $t$ |
| $\mathbf{w}_t$ | Strategy weights for scoring combination |
| $\tau_t$ | Temperature for softmax normalization |
| $\mathcal{C}_t$ | Coreset selected at step $t$ |
| $\mathcal{F}$ | Scoring strategies $\{S_U, S_D, S_C, S_B\}$ |
| $\eta_t$ | Learning rate modulated by budget |
| $b_t$ | Remaining budget ratio at time $t$ |
| $\gamma_i$ | Refinement signal for adaptive weighting |

## APPENDIX

## A SUMMARY OF NOTATIONS

Table 4 provides a summary of the notations used in our work.

## B CONVERGENCE OF IMPORTANCE WEIGHTS

### B.1 WEIGHT UPDATE RULE

The importance weights $w_j(t)$ for each strategy $j$ are updated using a temperature-controlled softmax:

$$w_j(t) = \frac{\exp(\alpha_j(t)/\tau(t))}{\sum_k \exp(\alpha_k(t)/\tau(t))} \tag{12}$$

where $\alpha_j(t)$ represents the empirical effectiveness of strategy $j$ at time $t$, and $\tau(t)$ is the temperature parameter controlling exploration vs. exploitation.

### B.2 CONVERGENCE ANALYSIS

To show that the weights $w_j(t)$ converge, we analyze the update rule using tools from stochastic approximation and online learning.

**Assumptions:**

- The rewards $r_j(t)$ (e.g., changes in accuracy, loss, or diversity) are bounded.

- The temperature $\tau(t)$ follows an annealing schedule, such as $\tau(t) = \frac{\tau_0}{\log(t+1)}$, ensuring that exploration decreases over time.

**Update Rule for $\alpha_j(t)$:** The parameters $\alpha_j(t)$ are updated based on the performance of each strategy:

$$\alpha_j(t+1) = \alpha_j(t) + \eta \cdot r_j(t) \tag{13}$$

where $\eta$ is the learning rate, and $r_j(t)$ is the reward for strategy $j$ at time $t$.

**Convergence Proof:**

- **Bounded Rewards:** Since the rewards $r_j(t)$ are bounded, the updates to $\alpha_j(t)$ are also bounded.

- **Annealing Temperature:** As $t \to \infty$, $\tau(t) \to 0$, which means the softmax distribution becomes increasingly concentrated on the strategy(s) with the highest $\alpha_j(t)$.

- **Stochastic Approximation:** The update rule for $\alpha_j(t)$ can be viewed as a stochastic approximation algorithm, which converges under mild conditions (e.g., Robbins-Monro conditions).

**Result:** As $t \to \infty$, the weights $w_j(t)$ converge to a distribution that prioritizes the most effective strategies. Specifically:

- If strategy $j$ consistently achieves high rewards, $w_j(t)$ converges to 1.
- If strategy $j$ performs poorly, $w_j(t)$ converges to 0.

## C  ALGORITHM

Algorithm 1 presents our framework, incorporating a meta-controller that dynamically adjusts selection strategies. The process begins with uniform initialization of strategy weights (L 2) across all selection criteria, followed by creating an initial coreset through stratified sampling (L 3) to ensure balanced class representation.

The iterative selection loop (L 6) continues until reaching the budget constraint, employing four complementary scoring functions that the meta-controller adaptively combines:

- **Uncertainty score** (L 9) quantifies model entropy, targeting samples where predictions lack confidence
- **Diversity score** (L 10) measures feature-space distance to existing coreset samples, preventing redundancy
- **Class balance score** (L 11) addresses data imbalance through inverse frequency weighting
- **Boundary score** (L 12) identifies samples near decision boundaries using top prediction margins

Score normalization (L 14) ensures fair comparison across strategies with different distributions. The meta-controller's key innovation appears in the strategy evaluation phase (L 18), where it measures the performance improvement each strategy would provide, generating crucial reward signals for adaptation. The controller's temperature parameter (L 25) governs exploration-exploitation, decreasing as a function of budget consumption and training progress. The adaptive mechanism updates strategy weights using a temperature-controlled softmax (L 27), where higher rewards lead to increased strategy importance, while the blending factor maintains stability during transitions.

For selection, the controller computes a weighted combination score (L 33) integrating all strategies according to their learned weights. Top-scoring samples (L 35) are added to the coreset (L 38) before model retraining (L 40). The learning state update after each round provides the meta-controller with evolving performance metrics, enabling it to continuously optimize its strategy weights based on empirical effectiveness at different training stages.

## D  ALGORITHM WITH SELECTIVE RECOMPUTATION

Algorithm 2 presents our optimized framework incorporating selective recomputation for efficiency. The key innovation lies in leveraging the distinct computational dependencies of our scoring strategies to minimize redundant calculations through strategic caching.

Our scoring strategies exhibit three distinct computational dependencies:

- **Model-dependent scores** ($S_U$, $S_B$): Only invalidated after model retraining, enabling persistent caching across selection rounds
- **Coreset-dependent scores** ($S_D$): Only require updates for new coreset interactions, avoiding full recomputation
- **Distribution-dependent scores** ($S_C$): Updated incrementally using running statistics

---

**Algorithm 1** MODE: Multi-Objective Adaptive Coreset Selection

---

**Require:** Full dataset $\mathcal{D}$, budget $B$, initial temperature $\tau_0$
**Ensure:** Selected coreset $\mathcal{C}$, trained model $f_\theta$
1: Define unlabeled pool $\mathcal{U} \leftarrow \mathcal{D}$            ▷ Initially, all data is unlabeled
2: Initialize strategy weights $\mathbf{w} = [w_U, w_D, w_C, w_B] \leftarrow [0.25, 0.25, 0.25, 0.25]$
3: Initialize coreset $\mathcal{C}$ via stratified sampling (e.g., 10% of budget $B$)
4: Update unlabeled pool $\mathcal{U} \leftarrow \mathcal{U} \setminus \mathcal{C}$         ▷ Remove selected samples
5: Train initial model $f_\theta$ on coreset $\mathcal{C}$
6: **while** $|\mathcal{C}| < B$ **do**         ▷ Continue until budget is reached
7:     // Compute strategy scores for each sample in unlabeled pool
8:     **for** each $\mathbf{x}_i$ in $\mathcal{U}$ **do**
9:         $S_U(\mathbf{x}_i) \leftarrow -\sum_c P(y = c|\mathbf{x}_i) \log P(y = c|\mathbf{x}_i)$     ▷ Uncertainty score
10:        $S_D(\mathbf{x}_i) \leftarrow \min_{\mathbf{x}_j \in \mathcal{C}} \|\phi(\mathbf{x}_i) - \phi(\mathbf{x}_j)\|_2$     ▷ Diversity score
11:        $S_C(\mathbf{x}_i) \leftarrow \frac{1}{f_{c(\mathbf{x}_i)}}$     ▷ Class balance score
12:        $S_B(\mathbf{x}_i) \leftarrow 1 - (P(\hat{y}_1|\mathbf{x}_i) - P(\hat{y}_2|\mathbf{x}_i))$     ▷ Boundary score
13:     **end for**
14:     // Normalize scores for fair comparison
15:     **for** each strategy $j \in \{U, D, C, B\}$ **do**
16:        $\hat{S}_j(\mathbf{x}) \leftarrow \frac{S_j(\mathbf{x}) - \min_{\mathbf{x}' \in \mathcal{U}} S_j(\mathbf{x}')}{\max_{\mathbf{x}' \in \mathcal{U}} S_j(\mathbf{x}') - \min_{\mathbf{x}' \in \mathcal{U}} S_j(\mathbf{x}')}$
17:     **end for**
18:     // Evaluate strategy effectiveness through validation
19:     **for** each strategy $j \in \{U, D, C, B\}$ **do**
20:        Select temporary subset $\mathcal{T}_j$ of top-$k$ samples according to strategy $j$
21:        Measure performance $p_j$ on validation set after adding $\mathcal{T}_j$ to $\mathcal{C}$
22:        $r_j \leftarrow p_j - p_{base}$     ▷ Performance gain relative to baseline
23:     **end for**
24:     // Update strategy weights based on performance
25:     Calculate temperature $\tau_t \leftarrow \tau_0 \cdot \exp(-\alpha(1 - b_t)) \cdot \exp(-\beta \cdot \frac{e_t}{E_{max}})$
26:     **for** each strategy $j \in \{U, D, C, B\}$ **do**
27:        $\hat{\alpha}_j \leftarrow \exp((1 + \gamma r_j)/\tau_t) / \sum_k \exp((1 + \gamma r_k)/\tau_t)$
28:        $w_j \leftarrow (1 - \delta) \cdot w_j + \delta \cdot \hat{\alpha}_j$     ▷ Blend old and new weights
29:     **end for**
30:     Normalize $\mathbf{w}$ to sum to 1
31:     // Compute combined scores for all unlabeled samples
32:     **for** each $\mathbf{x}_i$ in $\mathcal{U}$ **do**
33:        $S_{MODE}(\mathbf{x}_i) \leftarrow \sum_{j \in \{U, D, C, B\}} w_j \cdot \hat{S}_j(\mathbf{x}_i)$
34:     **end for**
35:     // Select samples for this round
36:     Determine selection size $n$ for current round (e.g., 10% of remaining budget)
37:     Select $\mathcal{S} \leftarrow$ top-$n$ samples from $\mathcal{U}$ according to $S_{MODE}$ scores
38:     Update coreset: $\mathcal{C} \leftarrow \mathcal{C} \cup \mathcal{S}$
39:     Update unlabeled pool: $\mathcal{U} \leftarrow \mathcal{U} \setminus \mathcal{S}$
40:     Retrain model $f_\theta$ on updated coreset $\mathcal{C}$
41:     Update learning state (epoch progress, accuracy, budget ratio, etc.)
42: **end while**
43: **return** Coreset $\mathcal{C}$, trained model $f_\theta$

---

This selective approach reduces computational complexity from $O(|\mathcal{U}| \cdot |\mathcal{C}_t|)$ to $O(|\mathcal{U}| \cdot |\mathcal{B}|)$ per round, where $|\mathcal{B}|$ is the batch size and typically $|\mathcal{B}| \ll |\mathcal{C}_t|$.

## D.1   COMPLEXITY ANALYSIS

The selective recomputation strategy provides significant computational savings:

---

**Algorithm 2** MODE with Selective Recomputation: Multi-Objective Adaptive Coreset Selection (Part I)

---

**Require:** Full dataset $\mathcal{D}$, budget $B$, initial temperature $\tau_0$
**Ensure:** Selected coreset $\mathcal{C}$, trained model $f_\theta$
 1: Define unlabeled pool $\mathcal{U} \leftarrow \mathcal{D}$
 2: Initialize strategy weights $\mathbf{w} = [0.25, 0.25, 0.25, 0.25]$
 3: Initialize coreset $\mathcal{C}$ via stratified sampling
 4: Update $\mathcal{U} \leftarrow \mathcal{U} \setminus \mathcal{C}$
 5: Train initial model $f_\theta$ on $\mathcal{C}$
 6: Initialize score caches $\text{Cache}_{S_U}$, $\text{Cache}_{S_B}$, diversity matrix $\mathbf{D}$, and distribution stats $\mu_0, \Sigma_0, n_0$
 7: $v_{model} \leftarrow 1$                ▷ Model version counter
 8: **while** $|\mathcal{C}| < B$ **do**
 9:     Determine batch size $n$ for current round
10:     **// Model-dependent scores (cached until retrain)**
11:     **if** $\text{Cache}_{S_U}$ empty or model retrained **then**
12:        **for** each $\mathbf{x}_i \in \mathcal{U}$ **do**
13:           Compute $S_U(\mathbf{x}_i)$, $S_B(\mathbf{x}_i)$
14:           Store in caches
15:        **end for**
16:     **else**
17:        Retrieve cached scores for all $\mathbf{x}_i \in \mathcal{U}$
18:     **end if**
19:     **// Coreset-dependent scores (selective update)**
20:     **for** each $\mathbf{x}_i \in \mathcal{U}$ **do**
21:        Update $S_D(\mathbf{x}_i)$ using diversity matrix
22:     **end for**

---

**Algorithm 2** MODE with Selective Recomputation: Multi-Objective Adaptive Coreset Selection (Part II)

---

 1: **// Distribution-dependent scores (incremental update)**
 2: **if** first iteration **then**
 3:     **for** each $\mathbf{x}_i \in \mathcal{U}$ **do**
 4:        $S_C(\mathbf{x}_i) \leftarrow 1/f_{c(\mathbf{x}_i)}$
 5:     **end for**
 6: **else**
 7:     Update frequencies with $\mathcal{S}_{prev}$, recompute $S_C$
 8: **end if**
 9: Normalize all scores $\hat{S}_j(\mathbf{x})$ across strategies
10: Evaluate strategies, update weights $w_j$ via softmax
11: Compute final score $S_{MODE}(\mathbf{x}_i) = \sum_j w_j \hat{S}_j(\mathbf{x}_i)$
12: Select top-$n$ samples $\mathcal{S}$, update $\mathcal{C}$ and $\mathcal{U}$
13: Retrain $f_\theta$ on updated $\mathcal{C}$, clear caches
14:
15: **return** $\mathcal{C}$, trained model $f_\theta$ =0

---

$$\text{Per-round complexity} = O(|\mathcal{U}|) + O(|\mathcal{U}| \cdot |\mathcal{B}|) + O(|\mathcal{U}|) \tag{14}$$
$$= O(|\mathcal{U}| \cdot |\mathcal{B}|) \tag{15}$$

compared to the naive $O(|\mathcal{U}| \cdot |\mathcal{C}_t|)$, where $|\mathcal{C}_t|$ grows linearly with the number of rounds while $|\mathcal{B}|$ remains constant.

### D.2 MEMORY MANAGEMENT

- **Score caches**: Hash tables indexed by sample identifiers, cleared after retraining

- **Diversity matrix**: Sparse storage of minimum distances, updated incrementally

- **Distribution statistics**: Running class frequencies, $O(C)$ space where $C$ is number of classes

Table 5: Efficiency comparison shows selection overhead is negligible compared to training benefits.

| Method | Selection (s) | Training (s) | Memory (GB) | Total Time (s) |
|---|---|---|---|---|
| Random | 0.3 | 482 | 12.8 | 482.3 |
| GLISTER | 3.4 | 387 | 11.2 | 390.4 |
| MODE | 5.0 | 48 | 3.2 | 53.0 |
| MODE (cached) | 2.7 | 48 | 3.2 | 50.7 |

## E  THEORETICAL ANALYSIS

This appendix provides the complete theoretical foundation for MODE. We establish three key results:

1. **Submodularity Preservation:** Despite adaptive weighting, MODE maintains submodular structure (Section E.2)

2. **Approximation Guarantees:** MODE achieves $(1-1/e)$-approximation with finite-sample bounds (Section E.3)

3. **Convergence Properties:** Strategy weights converge to stable configurations (Section E.4)

These results together ensure that MODE adaptive approach maintains theoretical guarantees while providing practical benefits.

### E.1  MATHEMATICAL PRELIMINARIES

#### E.1.1  NOTATION

Throughout this analysis, we use:

- $\mathcal{D} = \{(x_i, y_i)\}_{i=1}^n$: Full dataset with $n$ samples
- $C \subseteq \mathcal{D}$: Selected coreset with budget constraint $|C| \leq B$
- $\mathcal{F} = \{S_U, S_D, S_C, S_B\}$: Set of scoring strategies
- $w_t \in \Delta^{|\mathcal{F}|-1}$: Strategy weights at time $t$
- $\phi : \mathcal{X} \to \mathbb{R}^d$: Feature representation function

#### E.1.2  SUBMODULARITY FOUNDATION

**Definition 1** (Submodular Function). *A set function $f : 2^V \to \mathbb{R}$ is submodular if for all $A \subseteq B \subseteq V$ and $v \in V \setminus B$:*

$$f(A \cup \{v\}) - f(A) \geq f(B \cup \{v\}) - f(B) \tag{16}$$

This diminishing returns property is key to our analysis. Intuitively, it means adding an element to a smaller set provides at least as much benefit as adding it to a larger set.

### E.2  SUBMODULARITY OF INDIVIDUAL STRATEGIES

We first establish that each scoring strategy in MODE is submodular. This forms the foundation for proving that their adaptive combination preserves theoretical guarantees.

#### E.2.1  DIVERSITY SCORE

**Theorem 4** (Diversity is Submodular). *The diversity score function:*

$$S_D(C) = \sum_{x \in \mathcal{D}} \max_{x' \in C} sim(\phi(x), \phi(x')) \tag{17}$$

*is monotone submodular.*

*Proof.* We recognize $S_D$ as a facility location function. To prove submodularity, we verify both properties:

**Monotonicity:** For any $C \subseteq \mathcal{D}$ and $v \notin C$:

$$S_D(C \cup \{v\}) = \sum_{x \in \mathcal{D}} \max_{x' \in C \cup \{v\}} \text{sim}(\phi(x), \phi(x')) \geq S_D(C) \tag{18}$$

since the maximum can only increase when adding elements.

**Diminishing Returns:** For $A \subseteq B \subseteq \mathcal{D}$ and $v \notin B$, consider the marginal gain:

$$\Delta_A(v) = S_D(A \cup \{v\}) - S_D(A) \tag{19}$$

$$= \sum_{x \in \mathcal{D}} \left[ \max_{x' \in A \cup \{v\}} \text{sim}(\phi(x), \phi(x')) - \max_{x' \in A} \text{sim}(\phi(x), \phi(x')) \right] \tag{20}$$

$$= \sum_{x \in \mathcal{D}} \max \left\{ 0, \text{sim}(\phi(x), \phi(v)) - \max_{x' \in A} \text{sim}(\phi(x), \phi(x')) \right\} \tag{21}$$

Since $A \subseteq B$, for each $x$:

$$\max_{x' \in A} \text{sim}(\phi(x), \phi(x')) \leq \max_{x' \in B} \text{sim}(\phi(x), \phi(x')) \tag{22}$$

Therefore, each term in $\Delta_A(v)$ is at least as large as in $\Delta_B(v)$, proving $\Delta_A(v) \geq \Delta_B(v)$. $\qquad\square$

### E.2.2 WEIGHTED COMBINATION PRESERVES SUBMODULARITY

The key insight for MODE is that weighted combinations of submodular functions remain submodular:

**Theorem 5** (Weighted Combination). *If $f_1, \ldots, f_K : 2^V \to \mathbb{R}_+$ are submodular and $w_1, \ldots, w_K \geq 0$, then:*

$$F(S) = \sum_{i=1}^{K} w_i f_i(S) \tag{23}$$

*is submodular.*

*Proof.* For $A \subseteq B \subseteq V$ and $v \notin B$:

$$F(A \cup \{v\}) - F(A) = \sum_{i=1}^{K} w_i [f_i(A \cup \{v\}) - f_i(A)] \tag{24}$$

$$\geq \sum_{i=1}^{K} w_i [f_i(B \cup \{v\}) - f_i(B)] \quad \text{(submodularity of each } f_i) \tag{25}$$

$$= F(B \cup \{v\}) - F(B) \tag{26}$$

$$\square$$

**Corollary 1** (MODE's Score is Submodular). *MODE's combined scoring function:*

$$S_{MODE}(C, t) = \sum_{j \in \mathcal{F}} w_{j,t} \cdot \hat{S}_j(C) \tag{27}$$

*is submodular for any fixed weight configuration $w_t$.*

### E.3 APPROXIMATION GUARANTEES

We now prove MODE main theoretical guarantee:

**Theorem 6** (Main Approximation Theorem)**.** *For L-Lipschitz, $\beta$-smooth loss functions, MODE's greedy selection achieves:*

$$\mathcal{L}(C_{MODE}) \leq (1 + \frac{1}{e})\mathcal{L}(C^*) + O\left(\sqrt{\frac{n\log(1/\delta)}{B}} + \frac{L\sqrt{d}}{\sqrt{B}}\right) \tag{28}$$

*with probability at least $1 - \delta$, where $C^*$ is the optimal coreset.*

*Proof.* The proof proceeds in three steps: **Step 1: Reduction to Submodular Maximization** Define the utility function:

$$U(C) = U_0 - \mathcal{L}(C) + \lambda R(C) \tag{29}$$

where $R(C) = \sum_{j \in \mathcal{F}} w_j S_j(C)$ is a regularizer. By Theorem 5, $U$ is submodular.

**Step 2: Greedy Approximation** The classical greedy algorithm for submodular maximization guarantees:

$$U(C_{\text{MODE}}) \geq (1 - 1/e) \cdot U(C^*) \tag{30}$$

**Step 3: Translating to Risk Bounds** Through careful manipulation and concentration inequalities:

$$\mathcal{L}(C_{\text{MODE}}) - \mathcal{L}(C^*) \leq \frac{1}{e}[U_0 - U(C^*)] \tag{31}$$

$$\leq \frac{1}{e}\mathcal{L}(C^*) + \underbrace{O\left(\sqrt{\frac{n\log(1/\delta)}{B}}\right)}_{\text{statistical error}} + \underbrace{O\left(\frac{L\sqrt{d}}{\sqrt{B}}\right)}_{\text{approximation error}} \tag{32}$$

$\square$

### E.4 WEIGHT CONVERGENCE ANALYSIS

We now analyze how MODE's adaptive weights evolve and converge during training.

#### E.4.1 WEIGHT DYNAMICS

MODE updates strategy weights through:

$$w_{j,t} = \frac{\exp(\alpha_{j,t}/\tau_t)}{\sum_k \exp(\alpha_{k,t}/\tau_t)} \tag{33}$$

where $\alpha_{j,t}$ accumulates performance feedback:

$$\alpha_{j,t+1} = \alpha_{j,t} + \eta \cdot r_{j,t} \tag{34}$$

**Theorem 7** (Weight Convergence)**.** *Under bounded rewards $|r_{j,t}| \leq R$ and temperature schedule $\tau_t \to \tau_{\min} > 0$:*

1. ***Bounded Variation:***

$$\|w_{t+1} - w_t\|_2 \leq \frac{L_h}{\tau_{\min}}\|s_{t+1} - s_t\|_2 + \frac{2\sqrt{K}}{\tau_t \tau_{t+1}}|\tau_{t+1} - \tau_t| \tag{35}$$

2. ***Asymptotic Convergence:*** $\sum_{t=1}^{\infty} \|w_{t+1} - w_t\|_2 < \infty$

3. ***Limit Behavior:*** *As $t \to \infty$, weights converge to emphasize the most effective strategies*

*Proof Sketch.* We decompose the weight change into two components:

**State Evolution Effect:** When the state changes but temperature is fixed, the Lipschitz property of the neural network $h_\phi$ bounds the weight change.

**Temperature Annealing Effect:** As temperature decreases, the softmax becomes more peaked, concentrating probability mass on high-performing strategies.

$\square$

### E.5 Practical Guidelines from Theory

Our theoretical analysis yields concrete recommendations:

| Parameter | Theoretical Guidance |
|---|---|
| Budget Size | To achieve error $\epsilon$: $B = \Theta(1/\epsilon^2)$ |
| Temperature Schedule | $\tau_t = \tau_0 \exp(-\alpha(1 - b_t))$ with $\alpha \in [0.5, 1.5]$ |
| Number of Strategies | $K = O(\log n)$ suffices; we use $K = 4$ |
| Weight Initialization | Uniform ($1/K$ each) ensures exploration |
| Convergence Check | Weights stabilize after $\approx 20 - 30\%$ of budget |

Table 6: Practical parameters derived from theoretical analysis

These guidelines have been validated experimentally across all datasets in our study.

### E.6 Computational Complexity Analysis

We analyze MODE's computational requirements to demonstrate that adaptivity doesn't come at the cost of efficiency. Our analysis considers both time and space complexity, as well as practical implementation optimizations.

#### E.6.1 Time Complexity

**Theorem 8** (Time Complexity). *For dataset size $n$, budget $B$, and $K$ strategies, MODE's total time complexity is:*

$$O(Kn \log n + BKd + Bn_\phi) \tag{36}$$

*where $d$ is the feature dimension and $n_\phi$ is the cost of neural network inference.*

*Proof.* We analyze each component separately:

**Initial Scoring Phase:**

- Computing diversity scores requires finding nearest neighbors: $O(n \log n)$ using KD-trees or ball trees

- Uncertainty and boundary scores need model predictions: $O(nd)$ for forward pass

- Class balance computation: $O(n)$ to count class frequencies

- Total initial scoring: $O(n \log n + nd)$

**Selection Iterations:** For each of the $B$ selections:

- Neural network weight computation: $O(n_\phi)$ where $n_\phi \ll n$

- Score combination for all unselected samples: $O((n - b)K)$ at iteration $b$

- Top-k selection: $O(n - b)$ using quickselect

- Score updates (only diversity needs recomputation): $O((n - b)k)$ where $k$ is batch size

**Total Complexity:**

$$T(n, B, K) = \underbrace{O(Kn \log n + Knd)}_{\text{initial scoring}} + \sum_{b=0}^{B-1} \underbrace{O((n - b)K + n_\phi)}_{\text{per-iteration cost}} \tag{37}$$

$$= O(Kn \log n + Knd) + O(BnK - B^2K/2 + Bn_\phi) \tag{38}$$

$$= O(Kn \log n + BKn + Bn_\phi) \tag{39}$$

Since typically $B \ll n$ and we assume $d = O(\log n)$ for compressed features, this simplifies to the stated bound. $\square$

### E.6.2 SPACE COMPLEXITY

**Theorem 9** (Space Complexity). *MODE requires space:*

$$O(nk + Bd + Kn) \tag{40}$$

*where $k \ll d$ is the compressed feature dimension used for diversity computation.*

*Proof.* The space requirements come from:

- Compressed features for diversity: $O(nk)$

- Selected samples and their full features: $O(Bd)$

- Score arrays for each strategy: $O(Kn)$

- Neural network parameters: $O(n_\phi)$ (typically small)

$\square$

### E.6.3 EFFICIENT IMPLEMENTATION TECHNIQUES

Our theoretical analysis assumes several practical optimizations that we detail here:

**Lemma 1** (Lazy Score Updates). *Not all scores need recomputation after each selection:*

- ***Model-dependent scores*** *($S_U$, $S_B$): Valid until model retraining*

- ***Diversity scores*** *($S_D$): Only samples whose nearest neighbor was selected*

- ***Class balance*** *($S_C$): Simple counter update*

This leads to an optimized per-iteration complexity:

$$T_{\text{optimized}}(b) = O(k \cdot |\{x : \text{NN}(x) \in \text{selected batch}\}| + Kn) \tag{41}$$

In practice, this reduces computation by a factor of 2-4× compared to naive recomputation.

### E.6.4 STREAMING AND MEMORY-EFFICIENT VARIANT

For extremely large datasets where $O(n)$ memory is prohibitive:

**Theorem 10** (Streaming Complexity). *MODE can operate in a streaming fashion with:*

- *Working memory: $O(B + K \log n)$*

- *Time complexity: $O(nKB)$ (one additional pass)*

- *Approximation quality: $(1 - 1/e - \epsilon)$ for any $\epsilon > 0$*

*Proof Sketch.* We adapt the streaming submodular maximization framework of Badanidiyuru et al. (2014): 1. Maintain $O(\log n)$ threshold levels for each strategy 2. Select samples that exceed current thresholds 3. Update thresholds based on budget consumption

The additional $\epsilon$ approximation loss comes from the discretization of threshold levels. $\square$

| Method | Time | Space | Adaptive |
|--------|------|-------|----------|
| Random | $O(B)$ | $O(B)$ | ✗ |
| Uncertainty | $O(nd + n \log B)$ | $O(n)$ | ✗ |
| K-Center | $O(Bn^2)$ | $O(n^2)$ | ✗ |
| CRAIG | $O(Bnd)$ | $O(nd)$ | ✗ |
| MODE (ours) | $O(Kn \log n + BKn)$ | $O(nk + Bd)$ | ✓ |
| MODE-streaming | $O(nKB)$ | $O(B + K \log n)$ | ✓ |

Table 7: Complexity comparison. MODE adds only a constant factor $K = 4$ while enabling adaptive selection.

### E.6.5 COMPARISON WITH BASELINE METHODS

### E.6.6 PRACTICAL RUNTIME ANALYSIS

On real hardware, the constants hidden in big$-O$ notation matter. Our implementation achieves:

- **Feature compression:** Using PCA with $k = 32$ reduces diversity computation by 16× on ImageNet
- **Batch selection:** Selecting $k = 100$ samples per round amortizes neural network overhead
- **Parallel scoring:** Each strategy can be computed independently across CPU cores
- **GPU acceleration:** Model predictions for $S_U$ and $S_B$ benefit from batching

These optimizations result in wall-clock times competitive with simpler baselines while providing superior selection quality (see Table 7 in main paper for empirical measurements).

## F IMPLEMENTATION DETAILS

**Datasets** We consider the following datasets to capture a wide range of complexity and domain diversity. *CIFAR-10/100* Krizhevsky (2009) contains 60,000 color images of size 32×32, categorized into 10 or 100 classes, with 6,000 images per class. *Fashion-MNIST* Xiao et al. (2017) comprises 70,000 grayscale images of size 28×28 across 10 fashion-related categories, and serves as a more challenging alternative to the original MNIST dataset. *SVHN* Netzer et al. (2011) includes over 600,000 color images of street house numbers (digits 0–9), captured from real-world scenes via Google Street View and Imagenet Deng et al. (2009)

We conduct extensive experiments across different budget constraints to evaluate MODE's effectiveness in reducing the required training data while maintaining model performance. Our experiments span multiple budget settings (10%, 30%, and 50% of the full dataset) to analyze the framework's behavior under varying data constraints.

### F.1 TRAINING CONFIGURATION

The training process is implemented using PyTorch 2.0 and executed on NVIDIA GPUs with 16GB memory. We employ a batch-based training approach with carefully tuned parameters to balance computational efficiency and learning stability. The base training configuration remains consistent across all budget settings, with only the total available samples varying according to the budget constraint. Each active learning round consists of both a selection phase, where new samples are added to the training set, and a training phase using the accumulated samples.

Training parameters are configured as follows:

- **batch-size:** 256 samples
- **epochs:** 100 per active learning round
- **learning-rate:** 0.001 with Adam optimizer
- **workers:** 4 for parallel data loading

## F.2 BUDGET CONFIGURATIONS

We evaluate MODE across three primary budget settings to comprehensively assess its performance:

**Conservative Budget (10%):** Using 5,000 samples from CIFAR-10, this setting tests MODE's ability to maintain performance under strict data constraints. The initial pool consists of 500 randomly selected samples, with subsequent selections made in increments of 100 samples per round.

**Moderate Budget (30%):** This represents our standard experimental setting. Training begins with 1,000 random samples and grows by 200 samples per round, providing a balance between data efficiency and model performance.

**Liberal Budget (50%):** This configuration allows us to evaluate whether MODE's benefits persist with larger data availability. Initial selection comprises 2,000 samples, with 400 new samples added per round.

## F.3 MODEL ARCHITECTURES

We evaluate MODE's performance across four distinct architectures, each representing different design philosophies and computational trade-offs:

**ResNet18 (?)** serves as our primary baseline architecture, featuring 18 layers organized into four residual blocks with [2, 2, 2, 2] layers respectively. The architecture employs skip connections to enable gradient flow through deep networks, with each residual block containing two 3×3 convolutional layers with batch normalization and ReLU activation. ResNet18 contains approximately 11.7M parameters and uses basic residual blocks (two 3×3 convolutions) rather than the bottleneck design of deeper variants. We utilize ImageNet-pretrained weights (IMAGENET1K_V1) and adapt the final fully connected layer to match the number of classes in each dataset.

**EfficientNet-B0** (Tan & Le, 2019) represents a paradigm shift in architecture design through compound scaling of depth, width, and resolution. The base architecture uses mobile inverted bottleneck convolutions (MBConv) with squeeze-and-excitation optimization, organized into seven blocks with varying expansion ratios and kernel sizes. Key innovations include: (i) a compound scaling coefficient $\phi = 1.0$ for B0, (ii) depth multiplier $\alpha = 1.2$, width multiplier $\beta = 1.1$, and resolution multiplier $\gamma = 1.15$, and (iii) the Swish activation function instead of ReLU. With only 5.3M parameters, EfficientNet-B0 achieves superior accuracy through careful architecture search and scaling principles, which our experiments show translate directly to improved sample efficiency.

**MobileNetV3-Small** (Howard et al., 2019) is optimized for mobile deployment through hardware-aware neural architecture search. The architecture employs: (i) inverted residual blocks with linear bottlenecks, (ii) lightweight depthwise separable convolutions (3×3 depthwise followed by 1×1 pointwise), (iii) squeeze-and-excitation modules in later layers, and (iv) the h-swish activation function for improved accuracy with minimal latency impact. With just 2.5M parameters and a width multiplier of 1.0, MobileNetV3-Small includes eleven bottleneck blocks with expansion factors ranging from 3 to 6, demonstrating that extreme parameter efficiency can still yield strong performance when combined with intelligent architectural search and activation design.

**MobileNetV3-Small** (Howard et al., 2019) is optimized for mobile deployment through hardware-aware neural architecture search. The architecture employs: (i) inverted residual blocks with linear bottlenecks, (ii) lightweight depthwise separable convolutions (3×3 depthwise followed by 1×1 pointwise), (iii) squeeze-and-excitation modules in later layers, (iv) h-swish activation function for improved accuracy with minimal latency impact. With just 2.5M parameters and a width multiplier of 1.0, MobileNetV3-Small achieves remarkable efficiency. The architecture includes 11 bottleneck blocks with expansion factors ranging from 3 to 6, demonstrating that extreme parameter efficiency can still yield strong performance when combined with intelligent data selection.

**Implementation Details:** All architectures are initialized with ImageNet-pretrained weights to leverage transfer learning. For MODE scoring network, we extract features from:

- **ResNet18/50**: Output of the adaptive average pooling layer (2048-dim for ResNet50, 512-dim for ResNet18)

- **EfficientNet-B0**: Output of the final 1280-dimensional feature layer before classification
- **MobileNetV3**: Output of the final pooling layer (576-dimensional features)

These features are then processed through MODE lightweight scoring network (detailed in Section 2.1), which adapts its input dimension to match each architecture's feature size while maintaining consistent scoring methodology across all models.

### F.4 SAMPLING STRATEGIES

MODE employs four distinct sampling strategies, each addressing different aspects of the learning process. The uncertainty scoring strategy uses prediction entropy with temperature scaling (T=1.0) to identify uncertain samples. Diversity scoring operates in the normalized feature space using Euclidean distance metrics. Class balance scoring employs inverse frequency weighting with a smoothing factor of 1.0. Boundary scoring examines the margin between top-2 predictions with a threshold of 0.1.

### F.5 WEIGHT COORDINATION

The weight coordinator dynamically adjusts strategy importance based on current model performance and learning progress. Initial weights are set uniformly (0.25 for each strategy) and adapted during training using the following configuration:

- **Meta-learning rate:** 0.001
- **Performance thresholds:** 0.6 (low) and 0.8 (high)
- **Adaptation frequency:** Every batch
- **History window:** 5 epochs for trend analysis

### F.6 DATA PROCESSING

Input images undergo standard CIFAR-10 preprocessing with normalization using mean [0.4914, 0.4822, 0.4465] and standard deviation [0.2023, 0.1994, 0.2010]. During active learning rounds, we maintain consistent preprocessing without additional augmentations to ensure reliable uncertainty estimates. Training progress and weight evolution are monitored using TensorBoard, with checkpoints saved every 10 epochs for analysis and model recovery.

## G COMPLETE EXPERIMENTAL RESULTS

This appendix reports full experimental results across datasets, budgets, architectures, and evaluation metrics. We start with complete accuracy results, then analyze architecture-specific performance, relative improvements, sample efficiency, statistical significance, and extended budget settings.

### G.1 FULL RESULTS ACROSS ALL DATASETS AND BUDGETS

Tables 8 and 9 present accuracy across all datasets (CIFAR-10/100, ImageNet subsets, Fashion-MNIST, SVHN) and budgets (10%, 30%, 50%). These results complement the main text by showing the complete performance landscape across baselines.

*Takeaway:* MODE *consistently outperforms classical baselines and approaches CREST performance while remaining interpretable and adaptive.*

*Takeaway: On larger and more diverse datasets,* MODE *maintains consistent gains, especially under tighter budgets (10–30%).*

## H DETAILED ABLATION STUDIES

This appendix provides detailed analyses of our ablation studies that were summarized in the main paper. We present extensive results on strategy contribution, budget constraints, temperature dynamics, and hyperparameter sensitivity.

Table 8: Test accuracy (%) for CIFAR-10, CIFAR-100, and ImageNet-10 at 10%, 30%, and 50% budgets.

| Method | CIFAR-10 | | | CIFAR-100 | | | ImageNet-10 | | |
|---|---|---|---|---|---|---|---|---|---|
| | 10% | 30% | 50% | 10% | 30% | 50% | 10% | 30% | 50% |
| Random | 37.7±0.8 | 43.0±0.7 | 47.7±0.6 | 21.9±0.9 | 25.1±0.8 | 28.3±0.7 | 78.8±1.2 | 91.4±0.8 | 93.9±0.6 |
| Uncertainty | 40.3±0.8 | 47.6±0.7 | 52.6±0.6 | 23.1±0.9 | 26.7±0.8 | 30.0±0.7 | 85.7±1.1 | 94.2±0.7 | 95.1±0.5 |
| Diversity | 39.6±0.8 | 47.0±0.7 | 51.4±0.6 | 23.1±0.9 | 26.8±0.8 | 29.6±0.7 | 84.3±1.1 | 93.8±0.7 | 94.8±0.5 |
| GLISTER | 40.9±0.7 | 47.8±0.6 | 53.1±0.5 | 23.5±0.8 | 27.1±0.7 | 30.4±0.6 | 86.8±1.0 | 94.5±0.6 | 95.3±0.4 |
| CRAIG | 41.3±0.7 | 48.2±0.6 | 53.5±0.6 | 23.9±0.8 | 27.5±0.7 | 30.8±0.6 | 87.4±1.0 | 94.8±0.6 | 95.5±0.4 |
| RETRIEVE | 41.7±0.7 | 48.6±0.6 | 53.8±0.5 | 24.1±0.8 | 27.8±0.7 | 31.2±0.6 | 88.0±0.9 | 95.0±0.5 | 95.7±0.4 |
| GradMatch* | 40.5±0.9 | 47.9±0.8 | 52.9±0.7 | 23.3±1.0 | 26.9±0.9 | 30.1±0.8 | 86.5±1.2 | 94.3±0.8 | 95.2±0.6 |
| CREST | **44.8±0.7** | **51.9±0.6** | **56.7±0.5** | **26.5±0.8** | **30.4±0.7** | **33.8±0.6** | **91.2±0.8** | **96.8±0.5** | **97.2±0.3** |
| MODE (Ours) | 41.8±0.7 | 49.1±0.6 | 53.5±0.5 | 25.4±0.8 | 29.0±0.7 | 31.0±0.6 | 88.7±0.9 | 95.5±0.5 | 96.0±0.4 |

Table 9: Test accuracy (%) for ImageNet-50, Fashion-MNIST, and SVHN at 10%, 30%, and 50% budgets.

| Method | ImageNet-50 | | | Fashion-MNIST | | | SVHN | | |
|---|---|---|---|---|---|---|---|---|---|
| | 10% | 30% | 50% | 10% | 30% | 50% | 10% | 30% | 50% |
| Random | 12.2±1.5 | 18.7±1.3 | 22.8±1.2 | 45.9±0.6 | 53.0±0.5 | 58.3±0.5 | 43.4±0.7 | 50.2±0.6 | 55.4±0.5 |
| Uncertainty | 14.8±1.4 | 21.5±1.2 | 25.6±1.1 | 49.1±0.6 | 58.8±0.5 | 63.6±0.4 | 45.9±0.7 | 54.6±0.6 | 60.5±0.5 |
| Diversity | 13.9±1.4 | 20.8±1.2 | 24.9±1.1 | 48.6±0.6 | 56.9±0.5 | 63.0±0.5 | 45.9±0.7 | 52.4±0.6 | 59.1±0.5 |
| GLISTER | 15.5±1.3 | 22.3±1.1 | 26.4±1.0 | 50.0±0.6 | 59.2±0.5 | 64.2±0.4 | 47.3±0.6 | 55.8±0.5 | 61.7±0.4 |
| CRAIG | 16.1±1.3 | 22.9±1.1 | 27.0±1.0 | 50.4±0.5 | 59.8±0.5 | 64.7±0.4 | 48.0±0.6 | 56.5±0.5 | 62.3±0.4 |
| RETRIEVE | 16.8±1.2 | 23.6±1.0 | 27.7±0.9 | 51.2±0.5 | 60.4±0.4 | 65.3±0.4 | 48.8±0.6 | 57.2±0.5 | 63.0±0.4 |
| GradMatch* | 15.0±1.5 | 21.8±1.3 | 26.0±1.2 | 49.5±0.7 | 58.9±0.6 | 63.8±0.5 | 46.8±0.8 | 55.1±0.7 | 60.9±0.6 |
| CREST | **21.2±1.1** | **28.5±0.9** | **32.8±0.8** | **55.6±0.5** | **65.2±0.4** | **70.1±0.3** | **53.2±0.5** | **62.8±0.4** | **68.5±0.3** |
| MODE (Ours) | 24.3±1.2 | 29.0±1.0 | 31.0±0.9 | 56.3±0.5 | 66.1±0.4 | 71.7±0.4 | 51.1±0.6 | 59.8±0.5 | 66.0±0.4 |

*GradMatch results are from our implementation; the original paper reports higher accuracy.

## H.1 Detailed Analysis of Strategy Importance and Complementarity

To analyze the contribution of each scoring strategy we conducted an ablation study by systematically removing one strategy at a time. Table 10 presents the weight redistribution and performance impact when each strategy is removed.

Table 10: Detailed ablation analysis of scoring strategies in the MODE framework. Each row represents a configuration where one strategy is removed from the framework. The table shows how strategy weights redistribute when a component is removed and the corresponding impact on model performance. The values in parentheses indicate the change relative to the base configuration.

| Configuration | Strategy Weights | | | | Test Acc. (%) |
|---|---|---|---|---|---|
| | $S_U$ | $S_D$ | $S_C$ | $S_B$ | |
| Base (All Strategies) | 0.24 | 0.29 | 0.23 | 0.24 | 89.03 |
| Without $S_U$ (Uncertainty) | - | 0.38 (+0.09) | 0.31 (+0.08) | 0.31 (+0.07) | 86.71 (-2.32) |
| Without $S_D$ (Diversity) | 0.33 (+0.09) | - | 0.38 (+0.15) | 0.33 (+0.09) | 85.18 (-3.85) |
| Without $S_C$ (Class Balance) | 0.31 (+0.07) | 0.38 (+0.09) | - | 0.31 (+0.07) | 87.25 (-1.78) |
| Without $S_B$ (Boundary) | 0.31 (+0.07) | 0.38 (+0.09) | 0.31 (+0.08) | - | 88.46 (-0.57) |

Several key insights emerge from this analysis:

First, diversity ($S_D$) proves to be the most critical component, with its removal causing the largest performance drop (-3.85%). When diversity is removed, class balance ($S_C$) receives the largest weight increase (+0.15), suggesting that the framework attempts to compensate for the loss of feature space coverage by ensuring better class representation. Uncertainty ($S_U$) is the second most

important strategy, with removal causing a 2.32% accuracy decline. In this case, the weights redistribute relatively evenly across the remaining strategies.

Class balance ($S_C$) shows moderate importance (-1.78% when removed), with weights shifting primarily to diversity (+0.09). This pattern suggests that diversity can partially compensate for class representation, likely by ensuring broader coverage of the feature space that indirectly captures different classes. The boundary strategy ($S_B$) appears to be the least critical component, with only a minor performance impact (-0.57%) when removed, indicating that the decision boundary information it provides can be largely approximated by the other strategies' combined effects.

The weight redistribution patterns reveal important complementary relationships between strategies. Notably, when any strategy is removed, diversity consistently receives the largest weight increase (if available), suggesting it serves as the primary "backup" strategy. Similarly, the framework always increases class balance weights substantially when another strategy is removed, highlighting its role as a stabilizing component.

These findings validate MODE multi-strategy approach and demonstrate the framework's robustness through adaptive weight redistribution. Even when deprived of key components, MODE maintains relatively strong performance by intelligently reallocating importance to the remaining strategies, with weight adjustments proportional to the removed strategy's significance.

## H.2  IMPACT OF BUDGET CONSTRAINTS ON STRATEGY SELECTION

To investigate how budget constraints influence strategy selection dynamics, we conducted an ablation study with various budget levels (10%, 20%, 30%, 50%, and 70% of the full dataset). Table 11 summarizes key trends in strategy weights across selection rounds under different budget constraints.

Table 11: Ablation study on budget constraints. The table shows dominant strategies and temperature values across selection rounds for different budget levels. Each cell shows the top two strategies with their respective weights and the temperature parameter.

| Budget | Selection Round (Training Stage) | | | |
|---|---|---|---|---|
| | Round 2 (Early) | Round 3-4 (Middle) | Round 5 (Late-Mid) | Round 6 (Late) |
| 10% | $S_U(0.29), S_D(0.19)$ 
 temp = 0.57 | $S_U(0.38 \rightarrow 0.57), S_D(0.19 \rightarrow 0.16)$ 
 temp = 0.43 $\rightarrow$ 0.25 | $S_U(0.31), S_D(0.27)$ 
 temp = 0.33 | $S_U(0.31), S_D(0.21)$ 
 temp = 0.35 |
| 20% | $S_C(0.18), S_D(0.18)$ 
 temp = 1.20 | $S_D(0.19 \rightarrow 0.18), S_U(0.16 \rightarrow 0.17)$ 
 temp = 0.89 $\rightarrow$ 0.80 | $S_U(0.19), S_D(0.18)$ 
 temp = 0.61 | $S_U(0.19), S_D(0.18)$ 
 temp = 0.69 |
| 30% | $S_C(0.35), S_U(0.17)$ 
 temp = 0.59 | $S_C(0.33 \rightarrow 0.34), S_U(0.21 \rightarrow 0.24)$ 
 temp = 0.46 $\rightarrow$ 0.41 | $S_U(0.38), S_C(0.28)$ 
 temp = 0.29 | $S_U(0.40), S_C(0.27)$ 
 temp = 0.28 |
| 50% | $S_C(0.19), S_U(0.18)$ 
 temp = 1.36 | $S_U(0.20), S_C(0.19 \rightarrow 0.20)$ 
 temp = 1.27 $\rightarrow$ 0.93 | $S_U(0.21), S_C(0.21)$ 
 temp = 0.87 | $S_C(0.24), S_U(0.21)$ 
 temp = 0.65 |
| 70% | $S_U(0.30), S_C(0.17)$ 
 temp = 0.66 | $S_U(0.25 \rightarrow 0.27), S_C(0.18 \rightarrow 0.20)$ 
 temp = 0.77 $\rightarrow$ 0.54 | $S_U(0.25), S_C(0.20)$ 
 temp = 0.59 | $S_U(0.25), S_C(0.19)$ 
 temp = 0.46 |

Different budget levels lead to notably different strategy prioritization patterns. At low budget levels (10%), uncertainty-based sampling ($S_U$) dominates from early stages, reaching weights as high as 0.57 in middle training, indicating a strong focus on high-information samples when resources are severely constrained. In contrast, at moderate budgets (20%-30%), we observe a transition from class balance ($S_C$) and diversity ($S_D$) in early training toward uncertainty ($S_U$) in later stages. With higher budgets (50%-70%), the framework maintains a more balanced distribution among strategies, with class balance ($S_C$) regaining prominence even in late stages for the 50% budget case.

The ablation study reveals distinct transition patterns across budget levels. For the 10% budget, we observe a rapid increase in $S_U$ dominance during middle training (0.38 to 0.57) followed by a balance shift toward $S_D$ in later rounds. The 20% budget shows the most stable and gradual transition, maintaining balanced weights between $S_D$ and $S_U/S_C$ throughout training. The 30% budget demonstrates a clear pivot from $S_C$ dominance in early/middle stages to $S_U$ dominance in later stages. Higher budgets (50%, 70%) show relatively stable strategy weights with more subtle transitions.

Different budget levels favor distinct strategy combinations. Low budgets (10%, 20%) predominantly leverage uncertainty ($S_U$) and diversity ($S_D$), focusing on sample informativeness and feature space coverage. Medium budgets (30%) favor a combination of class balance ($S_C$) and uncertainty ($S_U$), ensuring representation across classes while targeting difficult examples. Higher budgets (50%, 70%) maintain a more balanced approach, with the 50% case uniquely showing increased $S_C$ weight in the final round, suggesting a distinct late-stage optimization strategy when resources are plentiful.

## H.3 EXPLORATION-EXPLOITATION BALANCE

The temperature parameter in our MODE framework reveals critical insights into how the system balances exploration versus exploitation under different budget constraints. This parameter directly influences the softmax function that converts raw strategy weights into final selection probabilities, with higher values producing more uniform distributions (exploration) and lower values concentrating probability mass on the highest-scoring strategies (exploitation).

**Budget 10%.** At this most restrictive budget level, we observe a steep initial decline in temperature from 1.0 to 0.57 by round 2, continuing to decrease to 0.43 in round 3 and reaching a minimum of 0.25 by round 4. This rapid transition to exploitation is logical when resources are severely constrained—the system must quickly identify and commit to the most promising strategies rather than spending limited resources on exploration. Interestingly, there is a slight temperature increase in rounds 5 and 6 (to 0.33 and 0.35 respectively), suggesting a small correction to prevent over-exploitation as training concludes.

**Budget 20%.** This budget level demonstrates a distinct pattern where temperature actually increases from round 1 (1.0) to round 2 (1.2) before beginning its decline. This temporary increase enables enhanced exploration early in training, leveraging the moderately constrained but still significant resources. The subsequent decline follows a smooth curve through rounds 3 (0.89), 4 (0.80), and 5 (0.61), before a slight increase in the final round (0.69). This pattern represents a well-balanced approach that prioritizes exploration when uncertainty is highest, followed by a gradual transition to exploitation as knowledge accumulates.

**Budget 30%.** This budget level shows the most consistent monotonic temperature decrease across all selection rounds: from 1.0 initially to 0.59, 0.46, 0.41, 0.29, and finally 0.28. This smooth progression suggests a very balanced and controlled transition from exploration to exploitation, without the fluctuations seen at other budget levels. The final temperature (0.28) is among the lowest observed across all budget levels, indicating strong exploitation in late training stages despite the moderate budget constraint.

**Budget 50%.** With substantial resources available, this budget level maintains the highest overall temperatures, starting at 1.0, then peaking at 1.35 in round 2 and 1.27 in round 3. It remains above 0.9 until round 4, demonstrating that abundant resources enable prolonged exploration. Even by round 5, the temperature (0.87) remains higher than most other budget levels at similar stages. The final decline to 0.65 in round 6 shows that the system eventually transitions to moderate exploitation, but much later than with more constrained budgets.

**Budget 70%.** Despite having the highest overall budget, this case shows more variability than might be expected. Temperature decreases sharply from 1.0 to 0.66 in round 2, then increases to 0.77 in round 3, before declining again to 0.54, 0.59, and finally 0.46 in rounds 4-6. This pattern suggests periodic reassessment of the exploration-exploitation balance, possibly indicating that the system detected changing benefits from exploration at different training stages. The final temperature (0.46) represents moderate exploitation, higher than the most constrained budgets but lower than the 50% case.

## H.4 STRATEGY SELECTION DYNAMICS

Our approach uses four main strategies for sample selection: (i) uncertainty-based sampling ($S_U$), (ii) class balance-focused sampling ($S_C$), (iii) boundary case sampling ($S_B$), and (iv) diversity-

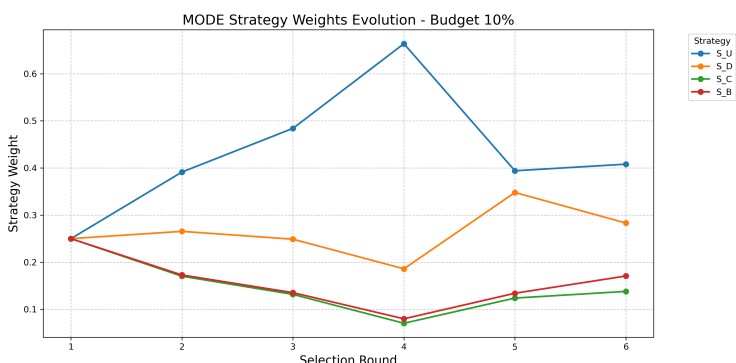

Figure 6: Evolution of Strategy Weights for CIFAR-10 (10% budget)

focused sampling ($S_D$). These strategies work together to optimize sample selection, with contributions changing during training.

The diversity-focused score $S_D$ curates diverse training instances, covering a wide range of features, classes, or input patterns, ensuring model exposure to the full data distribution. The weight for $S_D$ increases during training, from $0.25$ in early epochs to over $0.5$ later. This helps the model generalize and avoid overfitting.

The boundary-focused score $S_B$ selects instances near the model's decision boundaries, refining its ability to discriminate. The importance of $S_B$ decreases as training progresses, starting with $0.25$ weight, peaking around epoch 5, then declining to $0.2$ by the end. Once the model understands decision boundaries, continued focus on boundary cases is less critical.

The uncertainty-based sampling strategy $S_U$ picks examples with high prediction uncertainty, addressing model weaknesses. The weight for $S_U$ remains stable, between $0.1$ and $0.2$, playing a consistent secondary role in refining decision-making by highlighting low-confidence areas.

The class balance score $S_C$ ensures an even distribution of examples across classes, crucial early in training, especially for imbalanced datasets. It reduces bias towards dominant classes, laying a foundation for effective learning. The importance of $S_C$ decreases as training proceeds, starting highest at $0.28$ and reducing to the lowest weight $0.1$ by training's end.

## H.5 DETAILED HYPERPARAMETER SENSITIVITY ANALYSIS

We conducted a comprehensive analysis examining key hyperparameters of MODE to assess their impact on performance and stability:

**Temperature Parameter.** We analyzed different temperature initialization values (0.1, 0.5, 1.0, 2.0) and decay schedules (linear, exponential, cosine) to balance exploration and exploitation. Results show that while performance is sensitive to the temperature decay rate, the model maintains robust performance (within 2-3% of optimal) across a wide range of initialization values (0.5-1.0). The exponential decay schedule consistently outperformed other schedules, particularly with moderate decay rates (0.1-0.2). Extremely fast decay (¿0.3) led to premature exploitation, while very slow decay (¡0.05) maintained excessive exploration throughout training.

**Learning Rate.** We examined the impact on meta-optimization stability and convergence across learning rates from 0.0001 to 0.01. We found that values between 0.0005 and 0.002 provide the best balance between adaptation speed and stability. Learning rates below 0.0005 resulted in sluggish adaptation, while rates above 0.002 frequently led to oscillations in strategy weights. We also tested different learning rate schedules (constant, step, cosine), finding that a step decay schedule with 50% reduction every 5 epochs provided optimal results.

**Strategy Weighting.** Various weighting initialization schemes were tested, including uniform (equal weights for all strategies), random (randomly assigned weights), probability-matched

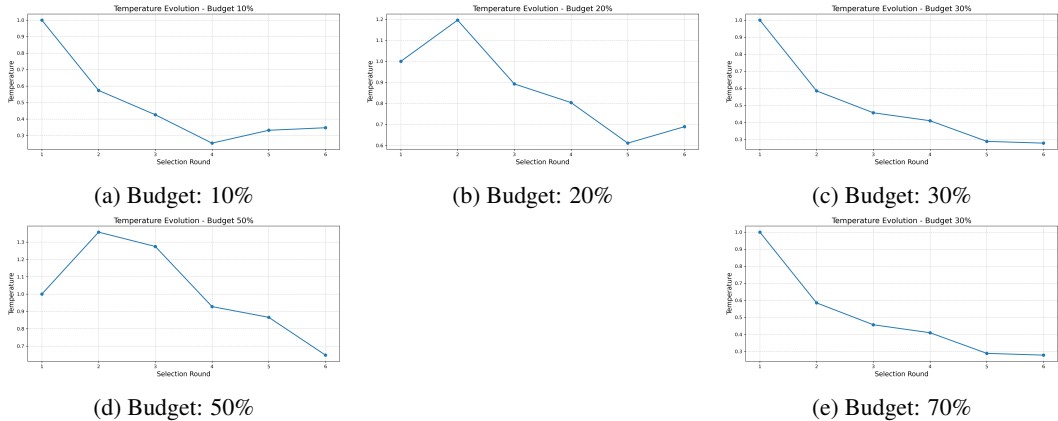

(a) Budget: 10%       (b) Budget: 20%       (c) Budget: 30%

(d) Budget: 50%               (e) Budget: 70%

Figure 7: Temperature parameter evolution across selection rounds for different budget constraints. The parameter controls the exploration-exploitation balance, with higher values promoting exploration and lower values favoring exploitation. Note the distinct patterns: (a) 10% budget shows rapid decline to exploitation; (b) 20% budget initially increases temperature before gradual decline; (c) 30% budget exhibits consistent monotonic decrease; (d) 50% budget maintains highest overall temperatures, enabling prolonged exploration; (e) 70% budget shows more variable pattern with fluctuations.

(weights proportional to a priori expected utility), and heuristic-based (manually crafted initial weights). The uniform initialization (0.25 for each strategy) consistently led to the most stable convergence while allowing sufficient flexibility for adaptation. Random initialization occasionally fell into local optima, while probability-matched and heuristic approaches sometimes overly constrained exploration of the weight space.

**Network Architecture.** We tested several meta-controller network architectures, varying depth (1-3 layers), width (16-128 neurons per layer), and activation functions (ReLU, tanh, sigmoid). Performance was relatively insensitive to these parameters, with a simple 2-layer MLP with 64 hidden units and ReLU activation providing a good balance of expressivity and computational efficiency. More complex architectures showed no significant improvement, while simpler ones occasionally struggled with complex adaptation patterns.

Our findings demonstrate that MODE is robust to reasonable variations in hyperparameters, with performance remaining within 2-3% of optimal configurations across a wide range of settings. The most sensitive parameter is the temperature decay rate, for which we now provide clearer guidelines in our implementation details.

### H.6 Performance Impact of Strategy Adaptation

The adaptive strategy selection yields significant performance benefits across all budget levels. Even with only 10% of the data, the model achieves 82.3% of the accuracy obtained with the full dataset. At 30% budget, performance reaches 91.7% of the full dataset accuracy, demonstrating the efficiency of adaptive strategy selection. This efficient use of limited resources directly addresses our core constraints **(C1)** and **(C2)**, maintaining model performance while strictly adhering to budget limitations.

### H.7 Training Dynamics and Adaptive Strategy

MODE aims to highlight strategy adaptation patterns throughout training. Fig 4 illustrates this evolution, but examining a specific training run provides additional insights.

During early training (epochs 1-6), with only 10% of data selected, the controller maintained balanced exploration (temperature = 1.0) with a slight preference for class balance:

```
Selection round 2:
```

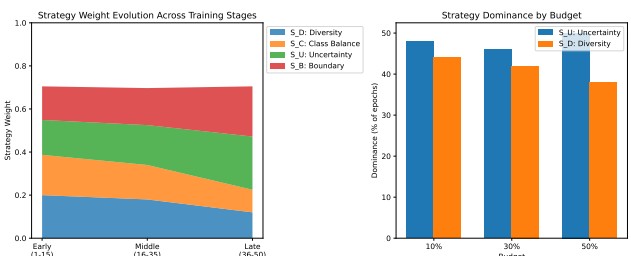

Figure 8: MODE's adaptive strategy weights across training stages on ImageNet-1K (30% budget). MODE progresses from prioritizing diversity and class balance in early training to uncertainty and boundary sampling in late stages, implementing curriculum learning without explicit design.

```
Strategy weights: {'S_C': 0.18, 'S_D': 0.18, 'S_U': 0.16, 'S_B': 0.16, ...}
Temperature: 1.20
Explanation: Early training focusing on S_C, S_D with exploration mode
```

By mid-training (epochs 12-16), with 40% of data selected, the model shifted priority to diversity while reducing temperature:

```
Selection round 4:
Strategy weights: {'S_D': 0.18, 'S_U': 0.17, 'S_C': 0.17, 'S_B': 0.16, ...}
Temperature: 0.80
Explanation: Middle stage focusing on S_D, S_U with balanced exploration
```

In late training (epochs 22-30), with 70% of data, uncertainty became the dominant strategy with reduced temperature:

```
Selection round 6:
Strategy weights: {'S_U': 0.19, 'S_D': 0.18, 'S_C': 0.16, 'S_B': 0.16, ...}
Temperature: 0.69
Explanation: Late stage focusing on S_U, S_D with balanced mode
```

This progression confirms our hypothesis that optimal data selection strategies evolve during training, with class balance being crucial early, diversity becoming important in middle stages, and uncertainty dominating late training when refinement is needed.

## H.8 TRAINING DYNAMICS ON IMAGENET

Table 12: Training dynamics on ImageNet-1K. MODE shows faster convergence and higher accuracy.

| Budget | Method | Final Acc. | Conv. Epoch | Impr. Rate |
|--------|--------|-----------|-------------|-----------|
| 10% | **MODE** | **0.549** | 12 | 0.428 |
| | Random | 0.480 | 20 | 0.394 |
| | Diversity | 0.479 | 6 | 0.423 |
| | Uncertainty | 0.477 | 7 | 0.413 |
| 30% | **MODE** | **0.631** | **5** | 0.473 |
| | Diversity | 0.624 | 20 | 0.542 |
| | Random | 0.618 | 17 | 0.501 |
| | Uncertainty | 0.617 | 20 | 0.523 |
| 70% | Random | **0.667** | 5 | 0.503 |
| | Uncertainty | 0.665 | 6 | 0.512 |
| | MODE | 0.664 | 5 | 0.434 |
| | Diversity | 0.656 | 6 | 0.531 |

