# OpenReview forum: "MODE: Multi-Objective Dynamic Coreset Selection"
_ICLR.cc/2026/Conference — ICLR 2026 Conference Withdrawn Submission_

### Official Review · Reviewer_hqLA · 2025-10-29

**Soundness:** 1
**Presentation:** 1
**Contribution:** 2
**Rating:** 2
**Confidence:** 5

**Summary:**

This paper proposes a new method for coreset selection. It progressively selects training data based on model training results. Several strategies are introduced to boost efficiency.

**Strengths:**

1. Although the performance is sub-optimal, extensive empirical experiments are conducted to compare MODE and baselines.

**Weaknesses:**

1. The title and method name are misleading. The problem formulation is not multi-objective in any sense. The problem statement minimizes only the coreset size as a single objective.
2. The paper is confusing and difficult to read. The description of the method—how the coreset is progressively selected—is missing from the main text. Section 2 merely lists several metrics and defers the full algorithmic details to the appendix, which severely disrupts readability. This issue recurs elsewhere (e.g., Line 265 refers readers to Table 12, which is located at the very end of the paper). Much of the main text focuses instead on implementation details related to efficiency optimization, which are too niche for the core paper. Section G.1, which contains important results, should be moved to the main body. The authors should reconsider the paper’s structure and prioritize what truly matters in the main text. Reviewers are not obliged to read the appendix, and the current layout makes the paper unnecessarily hard to follow.
3. Performance is sub-optimal. The method consistently underperforms compared to CREST. Furthermore, there is no efficiency comparison between MODE and CREST. If MODE takes longer to train while achieving worse results, its practical value is questionable.
4. Insufficient discussion of the method’s nature. The idea of progressively selecting training data during training closely resembles active learning, yet the paper barely discusses this connection. The authors attempt to overcome the one-shot limitation of traditional coreset selection by adopting a dynamic selection process inspired by active learning, where data are chosen iteratively throughout training. However, the proposed approach **retrains the model from scratch** at each selection stage, which contradicts the claim of being “adaptive” or “real-time.” According to Algorithm 1 (Line 40), the model is completely reinitialized after every validation step, so it does not retain or update knowledge adaptively across rounds.
5. Figures require major improvement. Figure 1 appears in the paper without any reference in the text, making its purpose unclear. Figure 2 has excessive white space, making it impossible to discern details without zooming in. Moreover, each subfigure is insufficiently discussed in the main text. I could continue listing such issues, but it would not be a productive use of time.
6. The proofs appear to omit important steps. For instance, the transition from Eq. (30) to Eq. (32) vaguely attributes the result to “careful manipulation and concentration inequalities” without explanation. Similarly, Eq. (35) presents only a proof sketch without a complete derivation. What's more, the theory does not seem justified by experiment results. In figure 2(d), I don't see how approximation scales as $O(\frac{1}{\sqrt{B}})$
7. The authors claim that their method provides an “interpretable” selection strategy, but no convincing evidence or analysis supports this statement.

**Questions:**

1. Table 5 requires further details. What is the experiment setting that arrives at this table? I couldn't find a description. You only compare with active learning baseline GLISTER. How does the total time to arrive at the coreset compare with other baselines, especially CREST?
2. In Figure 2, how does randomly select coresets perform in the $\epsilon$-approximation?

---

### Official Review · Reviewer_A2Wh · 2025-10-30

**Soundness:** 3
**Presentation:** 3
**Contribution:** 3
**Rating:** 6
**Confidence:** 2

**Summary:**

This paper investigates how to make data selection an intelligent process that adapts to a model’s learning state, transforming data selection from a static, one-size-fits-all decision into a dynamic, adaptive procedure. The core motivation stems from deep learning’s reliance on large-scale datasets and the substantial costs they incur. To address this challenge, a dynamic, multi-objective adaptive core-set selection framework, MODE, is proposed. MODE coordinates three key components: a set of complementary scoring strategies for evaluating sample value along multiple criteria (uncertainty, diversity, class balance, and boundaryness); an adaptive meta-controller that learns the optimal mixture of these strategies based on real-time training feedback; and a temperature-controlled weighting mechanism that balances exploration for the evolving model with exploitation of validated selection patterns. By allowing the meta-controller to dynamically adjust the emphasis of data selection according to the model’s learning state, MODE constructs compact, high-quality data subsets that substantially reduce training cost while preserving strong model performance, demonstrating superior data efficiency across multiple benchmark datasets.

**Strengths:**

1. The method is innovative, notably in the design of the adaptive meta-controller and the emergent curriculum learning. The meta-controller dynamically monitors training state and adjusts the weights of multiple data-selection strategies so that data selection aligns with the model’s evolving learning needs.

2. The experimental design is rigorous and well-justified. The authors compare MODE extensively to classic baselines such as random sampling and to recent benchmarks including CREST across datasets of different characteristics and scales, convincingly demonstrating MODE’s competitiveness. Detailed ablation studies and dynamic analyses further illuminate the internal mechanisms of the framework.

**Weaknesses:**

Major concerns:
1. In the baseline comparisons MODE’s accuracy is slightly lower than CREST, although MODE offers interpretable selection strategies. However, the interpretability argument currently stops at “what is selected” rather than “why that matters.” The paper shows (Figures 2 and 8) that early-stage selection emphasizes class balance and diversity while later stages emphasize uncertainty and boundary samples. While this is a useful observation, the manuscript does not validate whether this interpretability yields practical benefits—for example, can these observations help practitioners diagnose training problems, or guide future dataset collection? Without such evidence, using interpretability to compensate for the performance gap versus CREST is unconvincing.

2. There is a mismatch between the theoretical guarantees and the practical algorithm. Theorem 1 argues that MODE attains a near-optimal approximation ratio (1 − 1/e) under the assumption that scoring rules used for each selection step are fixed. However, Algorithm 1 shows that the scoring rules are temporarily modified during each selection based on exploratory results—i.e., the scoring rule is dynamic rather than fixed. Because the theoretical guarantee depends on a fixed scoring rule, it does not actually apply to the implemented, dynamically adapting algorithm. The authors should reconcile this gap and clarify whether (and under what conditions) the theoretical bound holds for the practical algorithm.

Minor revisions:
1. In Section F.3 (MODEL ARCHITECTURES), the description of MobileNetV3-Small appears twice.
2. In Section F.3 (MODEL ARCHITECTURES), the citation for ResNet-18 is currently a placeholder and should be replaced with the correct reference.

**Questions:**

1. How do the authors justify the near-optimal approximation bound given that the actual algorithm dynamically adjusts its scoring rules? Under what specific conditions does the theoretical guarantee remain valid for the implemented version of MODE?

2. If the theoretical bound does not directly apply to the adaptive setting, could the authors outline a potential extension or empirical validation to bridge this theoretical–practical gap?

---

### Official Review · Reviewer_Abdq · 2025-10-31

**Soundness:** 1
**Presentation:** 2
**Contribution:** 1
**Rating:** 2
**Confidence:** 4

**Summary:**

MODE is an adaptive core set selection framework that optimizes sample selection at different training stages by dynamically adjusting the weights of four sample selection strategies. This method has verified its effectiveness on multiple image classification datasets, especially outperforming traditional static methods in low-budget (10%-30%) scenarios. The paper also provides theoretical guarantees (such as approximation ratio and weight convergence) and detailed experimental analysis.

**Strengths:**

1. The paper conducts detailed theoretical derivations to elaborate on the proposed method.

2. The paper provides code for verifying the experimental results, and this practice is commendable.

**Weaknesses:**

1. The color in Figure 1 is so dull and unattractive.

2. Figure 2 is very small and unattractive.

3. The organizational structure of the thesis is very poor.

4. The latest baseline is CREST and it is from 2023. Surprisingly, its performance is even worse than that of the baseline from two years ago.

**Questions:**

1. What proportion of the paper is generated by large language models?

---

### Note · Authors · 2025-12-12

I have read and agree with the venue's withdrawal policy on behalf of myself and my co-authors.